# Unravelling the long-term, locally-heterogenous response of Greenland glaciers observed in archival photography

Michael A. Cooper[1], Paulina Lewińska[2], William A. P. Smith[2], Edwin R. Hancock[2], Julian A. Dowdeswell[3], and David M. Rippin[1]

[1]Department of Environment and Geography, University of York, York, UK
[2]Department of Computer Science, University of York, York, UK
[3]Scott Polar Research Institute, University of Cambridge, Cambridge, UK

**Correspondence:** David Rippin (david.rippin@york.ac.uk)

**Abstract.** We present an approach for extracting quantifiable information from archival aerial photographs to extend the temporal record of change over a region of the central eastern Greenland Ice Sheet. The photographs we use were gathered in the 1930s as part of a surveying expedition, and so they were not acquired with photogrammetric analysis in mind. Nevertheless, we are able to make opportunistic use of this imagery, as well as additional, novel data-sets, to explore changes at ice margins well before the advent of conventional satellite technology. The insights that a longer record of ice margin change bring is crucial for improving our understanding of how glaciers are responding to the changing climate. In addition, our work focuses on a series of relatively small and little studied outlet glaciers from the eastern margin of the Ice Sheet. We show that whilst air and sea surface temperatures are important controls on the rates at which these ice masses change, there is also significant heterogeneity in their responses, with non-climatic controls (such as the role of bathymetry in front of calving margins) being extremely important. In general, there is often a tendency to focus either on changes of the Greenland Ice Sheet as a whole, or on regional variations. Here, we suggest that even this approach masks important variability, and full understanding of the behaviour and response of the Ice Sheet requires us to consider changes that are taking place at the scale of individual glaciers.

## 1 Introduction

The Greenland Ice Sheet (GrIS) is the world's second largest ice mass and contains enough fresh water to raise sea level by $7.2\,\mathrm{m}$ (Hofer et al., 2020). Two decades ago, the GrIS was considered to exist in a state of quasi-stability with its regional climate, but recent climatic warming trends have resulted in it becoming by far the largest contributor (up to $1.2\,\mathrm{mm\,a^{-1}}$) to global sea level rise (Hanna et al., 2012; Van den Broeke et al., 2016). Between 1992 and 2018, $(3902 \pm 342)\,\mathrm{Gt}$ of ice was lost from the GrIS (Shepherd and IMBIE Team, 2020), but this has accelerated to an annual loss of $375\,\mathrm{Gt\,a^{-1}}$ of ice (on average) in the last decade (Enderlin et al., 2014; Van den Broeke et al., 2016). In the current year (2021), surface melting across large parts of the southern and coastal regions of the GrIS has been observed, with 2021 being the joint 14th highest melt year to date, with volumes substantially greater than the 1981-2021 average (http://nsidc.org/greenland-today/). The most recent publication by the Intergovernmental Panel on Climate Change (IPCC) reported that it is very likely that the Arctic has warmed at a rate that is more than twice that experienced globally over the past 50 years, and that it is virtually certain

that future warming will be greater than the global average (Masson-Delmotte et al., 2021). This accelerated retreat is a direct response to climatic warming, with an increase in mean surface air temperatures of $0.8\,°C$ between 2001 and 2011 (Hanna et al., 2012). These temperatures are significantly higher than any period in the last 100 years (Hanna et al., 2012).

The losses that are driven by these increased temperatures take place as both an increase in melting (i.e. surface mass balance changes) and an increase in ice discharge (e.g. Enderlin et al., 2014; Van den Broeke et al., 2016; Wood et al., 2021). Precise estimates of the relative contribution of each component vary, but Mouginot et al. (2019) estimate that over the 46 year period between 1972 and 2018, glacier dynamic processes contributed $66 \pm 8\%$ to mass loss, with surface mass balance changes constituting $34 \pm 8\%$. In recent years though, an increasingly negative mass balance has taken on a greater contribution to mass loss (Wood et al., 2021). Of the important dynamic processes, almost all of the increased ice discharge is considered to have come about through the retreat of ice fronts rather than processes that take place inland within the ice sheet (King et al., 2020). This has been partially attributed to warming ocean waters and increased surface runoff which result in a destabilisation of the marine termini and thus increased retreat rate and ice flow acceleration (Howat et al., 2008; Moon and Joughin, 2008; Seale et al., 2011; Murray et al., 2015; Wood et al., 2018, 2021).

Investigations of the GrIS and the changes that have gone on there are now extensive and well-documented (e.g. Enderlin et al., 2014; Van den Broeke et al., 2016; Goelzer et al., 2020; Hofer et al., 2020; Shepherd and IMBIE Team, 2020). Such investigations offer ice sheet-wide assessments of change which are regularly revised and updated. There are also multiple studies that are more focussed on individual ice streams and outlet glaciers, but such research tends to focus on the largest and most intensively investigated of these outlet glaciers, particularly, Jakobshavn Isbræ, Kangerlussuaq Glacier, and Helheim Glacier (Khan et al., 2020). Until recently there had been very few focussed studies of many of the hundreds of other smaller Greenlandic outlet glaciers, and so based on ice sheet wide analyses, it had been tempting to infer that change in this multitude of relatively poorly studied smaller outlets was homogeneous, in line with wider ice sheet behaviour (Moon et al., 2020). This, however, is a significant oversight since ice dynamics (and the now recognised importance of such dynamic processes in mass loss) makes studying and understanding the heterogeneous behaviour of all GrIS glaciers vital (King et al., 2020).

This is, of course, important because glaciers are key indicators of a changing climate (Haeberli, 2000; Holmlund et al., 2005). In direct response to global warming, the vast majority of the world's ice masses are in retreat (IPCC, 2014) and there is very high confidence that melting will continue for decades or centuries (IPCC, in press). However, despite continuing improvements in the understanding of the links between climate warming and the cryospheric response, there are still uncertainties surrounding the precise relationship between changes in an individual glacier area and volume and the climatic forces which drive them. This largely arises because glaciers are complex and because of a lack of available data. Greater understanding arises from longer-term time series of data from more ice bodies, and so, consequently, developing and expanding such data-sets on changing glacier dimensions, both spatially and temporally, is a key objective of much glaciological research, and arguably nowhere is this more important than in Greenland.

In this respect, satellite technology (Raup et al., 2006) has proved to be an extremely important and powerful tool for the monitoring and measurement of glacier change. Since the launch of the first Earth Resources Technology Satellite (ERTS1; now known as Landsat-1) in 1972 (Ives, 2011), it has been possible to use satellites to regularly track the changes experienced

in the cryosphere. Such abilities are important in our efforts to investigate the links between climate and glacier change, and variations in how glaciers respond to climate change across the world. Despite the undeniable value of satellite observations, and the insights they have afforded into cryospheric change, the period prior to the launch of ERTS1 is characterised by relative data sparseness (Goliber et al., 2021). In light of this, significant advances would be gained from further expansion of the record of glacier change into the past.

Here, we exploit a series of images gathered obliquely along the east coast of Greenland between Kangerdlugssuak and Umivik, along a c. 260 km-long section of coastline between 66.3° and 68.4°N, taken for surveying purposes. These images were gathered between 1930 and 1931 as part of the British Arctic Air Route Expedition (BAARE), which was carried out in an effort to discover the possibility of a new and shorter transit route between the UK and Canada. This route, in part, passes over Greenland, and one of the mission's aims was to survey the eastern and central parts of Greenland - the section of the proposed route that was least well known. The BAARE survey team did this using a ship and sea-plane, in an effort to photograph and map the coastline (The SPRI Picture Library, 1999).

The imagery that we utilize here provides oblique views of a number of outlet glaciers in two nearby regions. The focus of our study is the opportunistic 'snapshot' that these images provide of the state of these glaciers during the BAARE survey in 1930 (Shepherd and IMBIE Team, 2020). In this paper, we utilize Structure from Motion (SfM) approaches to build georeferenced orthophotos of the terrain within the BAARE imagery, so as to extract information about glacier extent from over 90 years ago. This provides an exceptional and important additional key to understanding ice mass change in this region way before the advent of satellite technology. To further supplement our investigations, we also add in additional steps between 1930 and the start of the Landsat record with orthophotos that we generate from imagery of the region from the now-declassified 1960s CORONA satellite mission (Shin, 2003), and similarly from aerial photography from the 1980s (Bjørk et al., 2012). Finally, we also explore the Landsat record from 1985 up to the present day. Overall, this suite of data provides unprecedented insights into the changes that have taken place over >90 years in this relatively poorly studied region of East Greenland, where important changes are nevertheless known to have taken place. To investigate this further, we also explore changes in air temperature, sea surface temperature and mass balance in an effort to identify the drivers of glacier change here.

## 1.1 Study area

Some of the imagery acquired during the BAARE expedition covered a part of Eastern Greenland that approximately corresponds to the 'Central Eastern Region' (CE) as defined by Mouginot et al. (2019) in their delineation of Greenland into 7 discrete regions. Other, more recent work, by King et al. (2020) divided the GrIS up into just 4 regions, with the BAARE sector under investigation here corresponding to the 'Southeast Region' (SE). This SE region is of great interest because whilst in all other parts of Greenland, glacier thinning is due (at least in part) to glacier discharge being greater than the balance flux (indicating dynamic disequilibrium), in this SE region, the primary cause of thinning prior to 2000 was increased surface melt, indicating that this region responds more rapidly to climatic forcing, reinforcing observations made by Hugonnet et al. (2021) of glacier fluctuations in agreement with precipitation and temperature fluctuations. Dividing the ice sheet into discrete regions like this has proved to be a powerful approach for exploring broad-scale patterns.

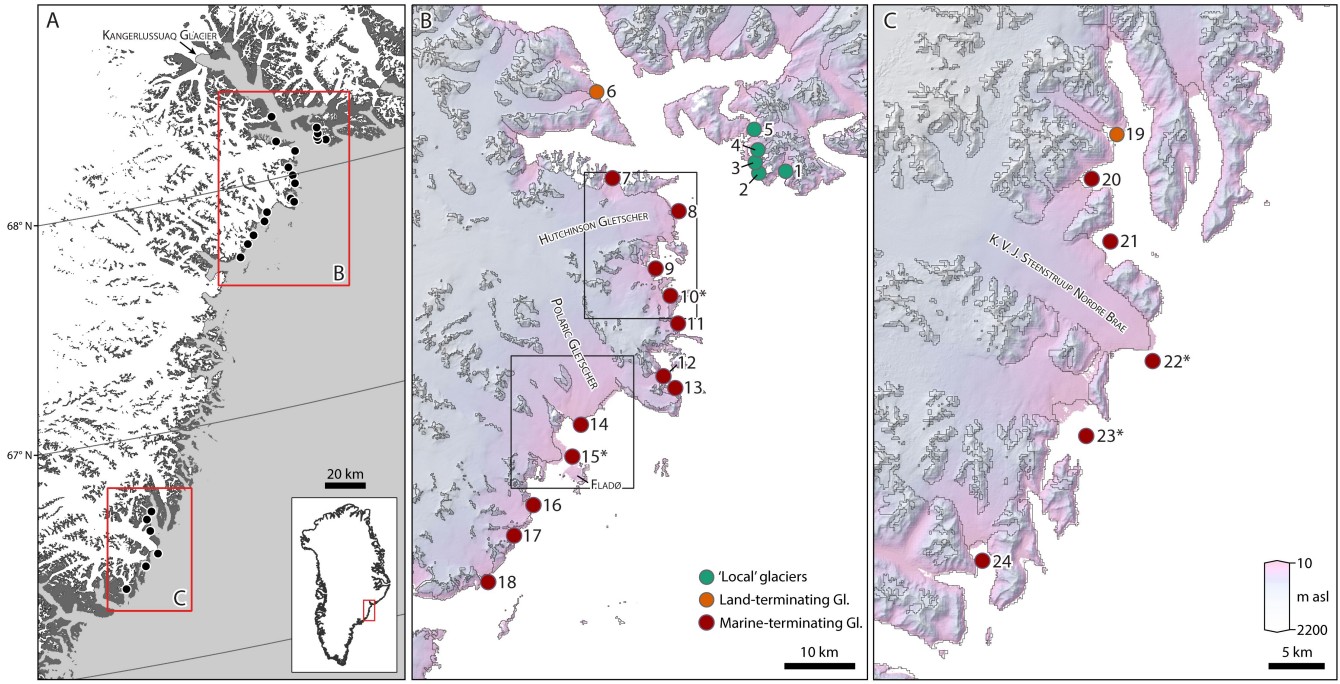

**Figure 1.** (A) Study location in East Greenland. Inset shows that the area of interest is in the central-eastern coastal region. Red boxes (labelled B and C) identify the northern and southern regions. The black dots within these boxes each represent the studied glaciers. (B) Northern study region incorporating 18 separate glaciers. Coloured circles indicate studied outlets, where the colour is indicative of the glacier type (see key). The largest outlets in this region are the Hutchinson Glacier (Glacier 8) and the Polaric Glacier (Glacier 14). Black boxes indicate further subdivisions of this area, discussed in the text. (C) Southern study region, incorporating 6 outlets, the largest of which is the KVJ Steenstruup Nordre Glacier (Glacier 22). Again, coloured dots represent glacier type, whilst the background in both B and C is shaded according to elevation (data provided by Bedmachine v3 (Morlighem et al., 2019)).

Figure 1 shows the study location in East Greenland. Whilst there are some large and well-studied outlet glaciers in this region (e.g. Kangerlussuaq (discharge in 2018 of $-162.5\,\mathrm{Gt\,a^{-1}}$; Mouginot et al. (2019)) and Helheim (discharge in 2018 of $-47.4\,\mathrm{Gt\,a^{-1}}$; Mouginot et al. (2019))), the area in general (and particularly its smaller glaciers) is relatively little studied. The biggest glaciers in the area are Hutchinson Gletscher, Polaric Gletscher and KVJ Steenstruup Nordre Brae, but the region is a mountainous and dense fjord and valley system and so glacier outflow is primarily dominated by relatively small outlet glaciers. The vast majority of the glaciers in our study are outlets of the GrIS but as shown in Figure 1, a small number are identified as being smaller local glaciers and ice caps (GICs) peripheral to the margins of the main GrIS (Rastner et al., 2012). As a consequence of the mountainous/alpine terrain, as well as its climatology, this area has much higher accumulation rates compared to the rest of the GrIS (and surface mass balance remains positive, which is discussed subsequently, cf. Figure 7c)).

## 2 Methods and materials

### 2.1 Archival photography

In order to retrieve historical information on the geometry of the 24 East Greenland glaciers we used three sets of photographic
data. Each of these sets was obtained using a significantly different imaging set-up, thus requiring customisation of the required processing methods.

*British Arctic Air Route Expedition, aerial oblique images - 1930-31*

The BAARE took place between July 1930 and August 1931. It was a privately funded expedition to investigate the feasibility of a new and shorter air passage between England and Canada. Part of the survey involved photogrammetric reconnaissance.
This was done with the use of two De Havilland DH.60 Moth planes with Gipsy 1 engines (Stephenson, 1932; Aviation Safety Network, 1999). One of the planes was equipped for taking vertical and oblique photographs (Watkins et al., 1932). A Williamson P14 camera with a lens of known focal length of $2209.8\,\mathrm{mm}$ (7.25'), and $127\,\mathrm{mm} \times 101.6\,\mathrm{mm}$ (5' $\times$ 4') glass plates with envelope adaptors for changing slides in daylight were used. Each flight took approximately 90 minutes and the plates were changed every 30 seconds. The time interval allowed for about 65% overlap on the photographs (Watkins, 1930).
During the summer of 1930 a total of 9 photographic flights (18h 20m) producing 450 plates were carried out. This covered the area from Bjorne Bugt up to and including Kangerdlugsuak Fjord, and also some parts of Sermilik Fjord and Angmagssalik Island. In the summer of 1931, due to poor weather conditions and the subsequent required dismantling and then rebuilding of each of the aircraft over winter and early spring, only two flights of 7 hours were carried out covering the area of Sermilik Fjord from Sermilik up to Umivik (Watkins et al., 1932). From all of this work, only 248 photographic plates remain, with scanned
versions held in the Picture Library of the Scott Polar Research Institute (The SPRI Picture Library, 1999). Unfortunately, the remaining images were lost after the expedition due to poor operational logistics by the returning party, as well as technical problems with the processing of the plates. Also some batches of plates were deemed to be unusable by cursory inspection and were subsequently destroyed. For our study we used 73 images obtained during the summers of 1930 and 1931.

*CORONA satellite mission, satellite stereo pair images - 1959-72*
The CORONA satellite mission was a clandestine surveillance program led by the CIA (Central Intelligence Agency) of the United States of America, and the U.S. Air Force, aimed at gathering spatial data for the creation of maps of vast remote areas for intelligence purposes (Goossens et al., 2006). Its existence was not acknowledged until the data entered the public domain in 1995. The CORONA data can currently be obtained (as digital high-resolution scans; $7\,\mathrm{\mu m}$) from the EarthExplorer website (U.S. Geological Survey USGS, 1995; Shin, 2003).
The CORONA mission was a vanguard in the early days of satellite surveillance. As such it piloted the use of sophisticated methods of shutter and camera construction, as well as an innovative and sometimes unreliable means of data retrieval. As a result, each mission had different operating specifications and a large amount of the data collected was not successfully retrieved. Generally speaking, for each mission the plan was to launch the satellite to a predetermined height, capture images on photographic film and then allow the satellite to return through the atmosphere and disintegrate on entry. Prior to this, a capsule

was jettisoned from the satellite and parachuted towards Earth containing the exposed film. This capsule was intercepted on descent by a plane (Galiatsatos, 2005). Since this was an intelligence collecting mission the capsule was designed to self-destruct if it was not intercepted before it reached a critical height. There was only one known occurrence when the self-destruction mechanism did not work as intended, and the capsule landed on the surface of the earth (Pieczonka et al., 2011).

In our study we used images taken on 24th September 1966. For this mission the KH-4M camera was used. This was an upgrade version of the KH-4 camera - the first stereoscopic camera used in space, providing 75% overlap. The KH-4A (Keyhole-4A) carried two J-1 (in earlier missions KH-3 cameras of $3.66\,\mathrm{m}$ resolution) panoramic cameras, with a focal length of $61\,\mathrm{cm}$, and a ground resolution of $2.7\,\mathrm{m}$ to $7.6\,\mathrm{m}$. It also carried a $4\,\mathrm{cm}$ index camera, with a focal length of $38\,\mathrm{mm}$, a ground resolution of $162\,\mathrm{m}$, and frame coverage of $308\,\mathrm{km} \times 308\,\mathrm{km}$. The J-1 cameras were placed on an M (Mural) mount, one pointing $15°$ aft from the vertical and the other $15°$ forward (Galiatsatos, 2005). The minimal flight height was $180\,\mathrm{km}$ and the duration of each mission was 14-15 days. Additional meta-data, such as ephemeris, ground velocity of the platform and the scan rate, the photographic coordinates of the principal points and the fiducial marks are not available (Shin, 2003).

The images obtained with the CORONA cameras have a complex image geometry (Casana and Cothren, 2015). The panoramic cameras used (also for aerial photography) work on the general principle that during the scanning process the lens and the scan arm moves while the film remains stationary. In this case the lens rotates around the second nodal point allowing the cylindrical focal plane to keep the image of distant objects sharp. As a result a 'bow-tie' shaped region is photographed and becomes compressed into a rectangular image. This effect creates significant panoramic image distortions. Additional significant imaging issues associated with those pictures are scan position distortion resulting from motion during the scanning process, image motion compensation distortion, tipped panoramic distortion, and geometric distortions resulting from roll, pitch, yaw and altitude instability. Many of these effects could be rectified with rigorous geometric distortions corrections, as is done with current satellite imaging systems. However, the lack of available meta-data makes such an approach intractable, thus necessitating a more customised approach (Shin, 2003; Galiatsatos, 2005).

*Greenland 1:15000 scale, vertical aerial images - 1978-87*

Aerial photographic missions were carried out between 1978-87 by the Geodetic Institute, the National Cadastre and Survey of Denmark, and the Danish Geodata Agency. More recently, these organisations were merged and renamed as the Agency for Data Supply and Efficiency (SDFE), which holds the records of the survey including the original photographs, scans, flight plans, and calibration data for both cameras. There is also GCP (Ground Control Point) data (obtained via triangulation, aero-triangulation and Doppler measurements) and this was used for the creation of a DEM model in the early 2010s (Bjørk et al., 2012). The photographic data covers all of Greenland together with the surrounding smaller islands, but excludes the interior of the ice sheet (Korsgaard et al., 2016). A WILD RC10 camera with a nominal focal length of $87.72\,\mathrm{mm}$ was used to collect super-wide-angle photographs at planned flying heights of $13\,000\,\mathrm{m}$. The images were captured on photographic film, in black and white and with 8 fiducial marks on each image. For our study we used 58 images obtained on 30th of July and 14th of August 1981. All were captured in favourable weather conditions and with at least 66% overlap between frames.

### 2.1.1 Geolocalisation and uncertainty of ground truth model

For the geolocalisation of the orthomosaic extracted from the archival images, the ArcticDEM (Morin et al., 2016) model was used. This is a relatively new product provided by the National Science Foundation (NSF) and the US National Geospatial-Intelligence Agency (NGA), which has been produced since 2015. The dataset is constructed by combining in-track and cross-track high-resolution (about $0.5\,\mathrm{m}$) imagery acquired using the DigitalGlobe constellation of stereoscopic optical imaging satellites, and including WorldView-1, WorldView-2, WorldView-3, and GeoEye (Meddens et al., 2018). It is created using Surface Extraction with the TIN-based Search-space Minimization (SETSM) algorithm (Noh and Howat, 2015). ArcticDEM raw products are additionally georeferenced by alignment to ICESat point cloud that has high $0.01\,\mathrm{m} \pm 0.07\,\mathrm{m}$ accuracy but coarse measurement footprint of $70\,\mathrm{m}$ (Morin et al., 2017). The current version of the ArcticDEM offers a highest resolution of $2\,\mathrm{m}$ raw data strips (day stamped) and a DSM mosaic averaged over time and area with a resolution of $2\,\mathrm{m}$.

There has been little research aimed at establishing the defined accuracy or consistency of these models. However, our own experiments and the results reported in Błaszczyk et al. (2019) both suggest that the $2\,\mathrm{m}$ day stamped strips are often wrongly aligned and prone to artefacts. These artefacts are usually 3D representations of cloud cover or random 'tower'-shaped elements (Crosby, 2016; Meddens et al., 2018). Moreover, the artefacts are hard to recognise in raster format due to a lack of corresponding texture, but become obvious after export to a point cloud format. Also, many of the strips are incomplete, having empty pixels. Thus it would be only possible to use them if combined with additional strips of the same area obtained on a different date (Barr et al., 2018). Lastly, the strips of adjoined areas have not all been captured at the same time. Combined with the movement of glacier front position and the changing pattern of snow cover could require us to reassemble the strips in order to obtain a unified model of the analysed area. Thus we decided to use the $2\,\mathrm{m}$ mosaic.

The mosaic was used as the basis for GCP (Ground Control Points). The GCP were chosen by comparing the archival images to shaded visualisations of the ArcticDEM mosaic and then identifying both the overlapping areas and easily identifiable points. Since neither the producer of the DSM nor the available scientific publications give definitive results of the ArcticDEM quality, the accuracy for our GCP was assumed to be the same as the pixel size of the ArcticDEM $2\,\mathrm{m}$ mosaic. It is also important to mention that we encountered areas on the mosaic that were clearly artefacts and we removed them from further analysis.

### 2.1.2 Structure-from-Motion based orthomosaics

Structure-from-Motion (SfM) has rapidly become one of the most popular means of obtaining 3D data from image sequences. Most SfM algorithms seek to simultaneously estimate a 3D scene model (sparse point clouds), camera intrinsic parameters (focal length, centre of projection etc.) and camera extrinsic parameters (3D pose, translation, rotation) from a set of overlapping images. This is aided, if needed, by geographical localisation information provided by GCP or camera path data. In general, SfM algorithms proceed by sequentially (a) extracting a set of distinctive local features in the available images, (b) robustly matching them across images, (c) optimising their 3D positions, (d) determining the camera parameters and (e) adding more images to the reconstruction during each iteration. The outputs of this process are the camera parameters and a sparse point cloud with 3D points consisting of matched 2D features (Ryan et al., 2015). Using fixed camera parameters so-obtained, a

| Time of acquisition | Type of images | Number of produced orthophotomaps | Number of used images | Total number of GCPs | Average number of GCPs per image | Number of GCPs per km$^2$ | Average 2D / 3D error [m] | Average orthophotomaps pixel size [m] |
|---|---|---|---|---|---|---|---|---|
| BAARE 1930/1931 | aerial oblique | 5 | 93 | 78 | 12 | 0.04 | 17.77 / 20.94 | 1.71 |
| CORONA 24.09.1966 | satellites stereo pair | 5 | 10 | 385 | 77 | 0.49 | 18.21 / 19.48 | 2.50 |
| 30.07.1981 14.08.1981 | aerial vertical | 4 | 30 | 168 | 18 | 0.04 | 21.50 / 25.58 | 2.24 |

**Table 1.** Archival Orthophotomaps: input data, quality and accuracy overview.

dense point cloud can be estimated using a process called Multi-View Stereo (MVS). This process is often based on performing dense binocular stereo between pairs of images with a large overlap. As a result multiple depth maps are effectively combined. A dense point cloud can be transformed into a mesh and then rendered with textures extracted from the original images (Park and Lee, 2019; Yurtseven et al., 2019).

The outputs of SfM can be used in a number of ways. One of the most relevant to the work reported here is the creation of an orthomosaic or orthophotomap. An orthomosaic is a combined image created by the seamless or near seamless merging of the original images projected onto the plane (or DEM/mesh model) and then transformed to the required projection. During this process the images are ortho-rectified (geometrically corrected) such that the scale is uniform and so that a photo or image adheres to a given map projection (Lamsters et al., 2020; Agisoft Metashape, 2020). In the work reported here, we

created orthophotomaps based on the images described in Section 2.1. This allowed us to create archival orthophotomaps and compare them to current Landsat-based satellite images (described in Section 2.2) thus providing detailed information on glacier front movements or overall glacier movement in the region studied. All of the datasets used the same projection, namely WGS84/NSIDC Sea Ice Polar Stereographic North (EPSG:3413). In order to georeference the archival orthophotomaps, a number of GCPs obtained from the ArcticDEM were used. This number varied with the size of the area constituting each

orthophotomap, but generally the 1960s dataset required the largest number of points due to its unique distortion properties. For each orthophotomap between 35 and 100 GCPs were used (Table 1) .

## 2.2 Archival orthophotomaps

The archival orthophotomaps were produced with the use of Agisoft Metashape (Agisoft Metashape, 2020). Since we did not have sufficient data to select the images with the best overlap or with the best light conditions, it was decided to use all the available images for this procedure. Initially we divided the 1930s data set into 5 regions for the production of 5 orthophotomaps. However, the 1960s and 1980s datasets were significantly different in terms of their extent and overlap. This forced us to divide our area into different sub-regions (Table 1) in order to produce mosaics of the same glaciers as covered by the 1930s images. In Figure 2 examples of typical source imagery are provided so that these can be compared. as well as derived SfM-based orthophotomaps.

Table 1 describes the data used and the accuracy of the results obtained from them. For all of the images we found the corresponding areas on the ArcticDEM model and created GCPs on stable, non-ice covered bare ground, which we assume to be fixed over the time period covered. The GCP placement accuracy calculated during the processing of the mosaics was around 1 pixel and in 90% of cases was smaller than 0.8 pixel. The spatial accuracy in metres varied, but in most cases was less than $20\,\mathrm{m}$, and did not exceed $15\,\mathrm{m}$ in the X and Y directions separately. This result can be considered satisfactory when taking into account the quality of the ground truth model, problems with the definition of the stable areas for GCPs, and the age and type of the archival images. We also considered isostatic uplift of GCPs over the 90 year period of our investigation. Shepherd and IMBIE Team (2020) explored rates of isostatic uplift rates over Greenland via a number of GIA models. On average, across our region of investigation, these models suggest uplift rates are approximately $0\,\mathrm{mm\,a^{-1}}$ to $2\,\mathrm{mm\,a^{-1}}$, which equates to a maximum potential mismatch of $18\,\mathrm{cm}$ over 90 years. In light of other much larger uncertainties, we do not consider this potential error source to be of significance. More detailed information on the creation of the orthophotomaps can be found in the supplementary materials.

## 2.3 Satellite imagery

In order to complete our record and extend the period covered by our study to the present day, we use satellite records to explore glacier change from 1985 to the present day. Imagery were downloaded from the USGS website EarthExplorer (U.S. Geological Survey USGS, 1995). We sought images with minimal cloud-cover that were gathered in July/August, when we expect surface snow cover to be at a minimum. We sourced imagery from Landsat-5 for the years 1985 and 1995, from Landsat-7 for the year 2005, and from Landsat-8 for the years 2015 and 2019 (all Landsat imagery courtesy of the U.S. Geological Survey). For each year for which we had data, composite images were generated.

## 2.4 Quantifying changing glacier margins

After the image processing was complete, seven discrete time steps of glacier extent were generated, covering a period of 89 years. For each year for which we had data, the margins were delineated manually. Prior to doing this, we investigated the use of semi-automatic margin detection approaches but found that a manual approach was more suitable and more accurate (Paul et al., 2017; Rippin et al., 2020). These delineated margins were then analysed to investigate glacier change over time.

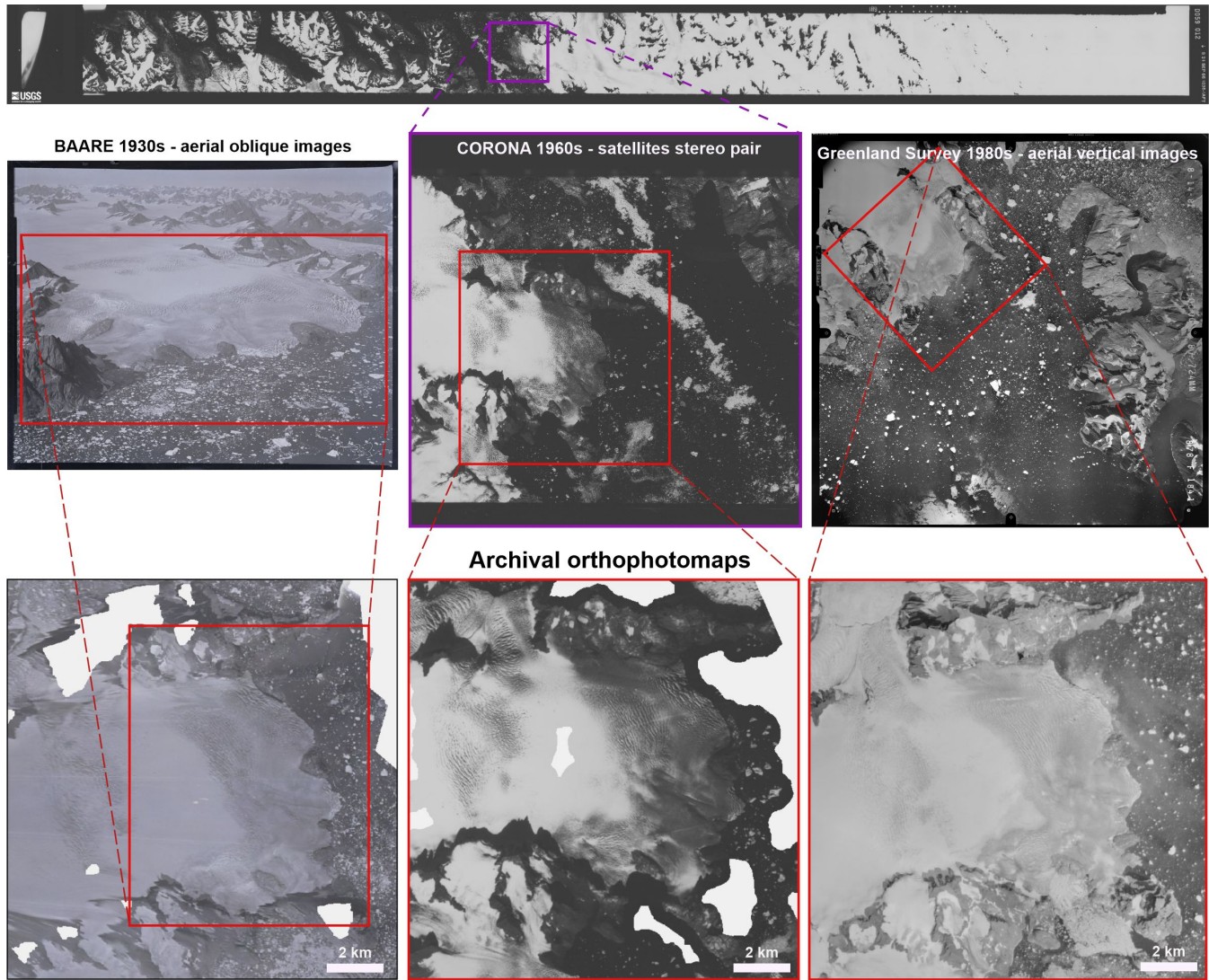

**Figure 2.** Illustration of typical source data and SfM-based orthophotomaps for archival image datasets. We show the BAARE (col. 1), CORONA (col. 2) and Greenland survey (col. 3) datasets. For BAARE and Greenland survey, we show source images in the second row. The CORONA images cover such a large area that we show a complete strip in the top row, then a region crop in the second. Our derived orthophotomaps are shown in the third row with the approximate correspondence to source images shown in red.

Specifically, we use contemporary velocity data to define a centre-line and measure a single length change at locations where the centre-line intersects delineated frontal margins. In order to normalise for the different duration of each timestep, we convert these distances to rates of change in units of metres per year. The only source of uncertainty in these measurements is the accuracy of the corresponding orthomosaics. The accuracy for each pixel in the BAARE, CORONA, 80s, Landsat 5 and Landsat 7/8 orthomosaics is respectively, GCP error $17.77\,\mathrm{m}$ with pixel size $1.71\,\mathrm{m}$, $18.21\,\mathrm{m}$ with pixel size $2.50\,\mathrm{m}$, $21.50\,\mathrm{m}$

with pixel size 2.24 m, 12 m with pixel size 30 m and 12 m with pixel size 15 m. Based on this information we calculate RMSE errors for distance measurements for time steps 30s-60s, 60s-80s, 80s-1995, 1995-2005, 2005-2015/2015-2019 as respectively 26 m, 30 m, 39 m, 37 m, 27 m.

## 2.5 Additional datasets

In addition to our focus on imagery as outlined above, subsequent investigations and analyses also make use of a range of additional environmental datasets. Here we briefly outline these and their sources.

### 2.5.1 Air temperature

Mean annual minimum and maximum air temperatures, as well as positive degree days (calculated from these data), were acquired from a meteorological station maintained by the Danish Meteorological Institute at Tasilaaq, close to our area of investigation. Temperature data were accessed via https://www.dmi.dk/fileadmin/Rapporter/2020/DMIRep20-04.pdf with the assistance of A. Bjørk (personal communication, January 2021). We compare temperatures and subsequent variables relative to a baseline originally defined by Box et al. (2009). Box et al. (2009) state that a period of thirty years (1951-1980) is generally considered to be long enough to be taken as a 'climate norm', against which anomalies can be determined. For consistency with Box et al. (2009), we define our baseline in precisely the same way.

### 2.5.2 Sea surface temperature (SST)

Mean annual sea surface temperature (SST) data were determined from the Hadley Centre Sea Ice and Sea Surface Temperature data set. This was taken from the UK Meteorological Office Marine Data Bank (MDB). See Rayner et al. (2003) and https://www.metoffice.gov.uk/hadobs/hadisst/index.html for a full explanation of the data sources.

### 2.5.3 Surface mass balance (SMB)

Direct measurements of surface mass balance data from this part of Greenland are very rare, and in fact are only available for any significant duration from a single glacier - Mittivakkat Gletscher (Mernild et al., 2011; Bjørk et al., 2012). It is this lack of mass balance data that makes exploring frontal change as a proxy for mass balance so important (Bjørk et al., 2012). Here we utilize modelled surface mass balance data following the approach of Wake et al. (2009) and Box and Colgan (2013) for South East Greenland and have explored how this varied over our study period. As with SST, mass balance fluctuations are displayed with reference to a baseline defined as the mean SMB over the period 1951 to 1980 (Box et al., 2009).

### 2.5.4 Offshore bathymetry

We utilised a small amount of bathymetric data from Kangerlussuaq Fjord. The data are in the British Antarctic Survey (BAS) database, but were obtained from the UK's RRS James Clark Ross on cruise JR106N (PI J.A. Dowdeswell) between 13th

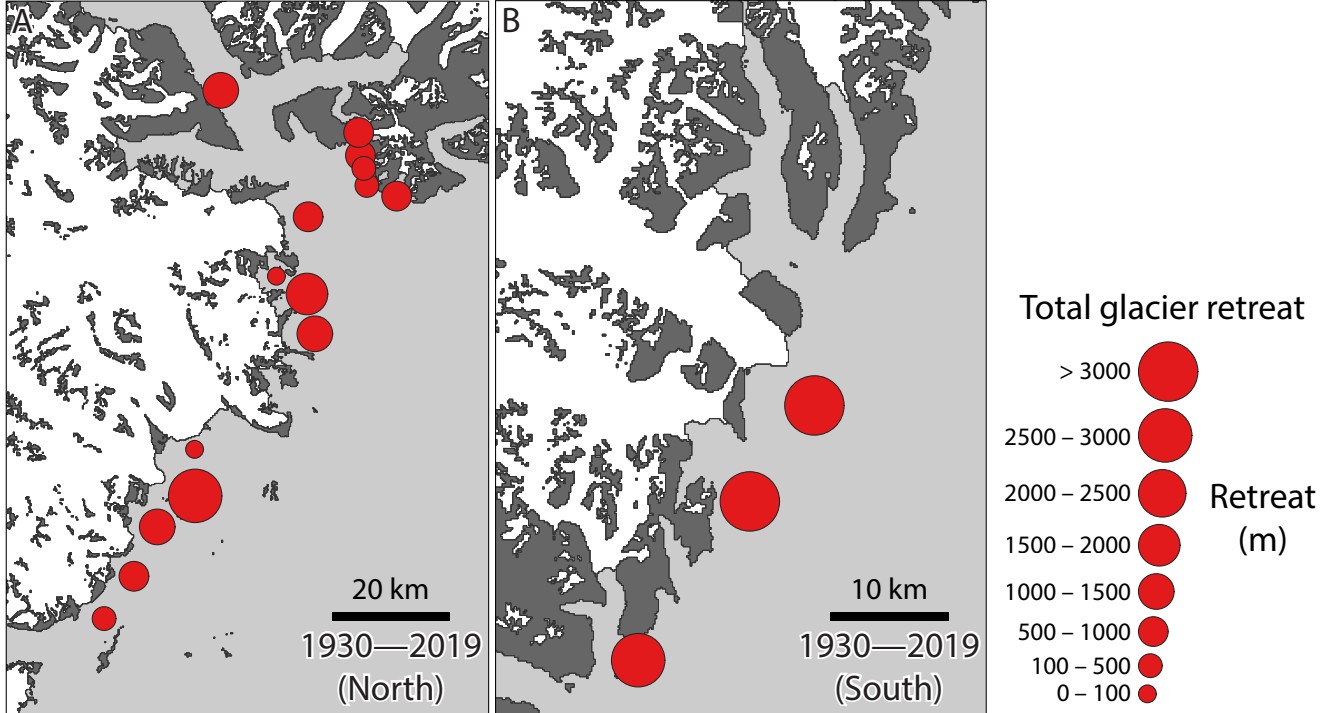

**Figure 3.** Graphical representation of overall change in glacier frontal position rate (1930-2019) in both the Northern (A) and Southern (B) study areas in East Greenland (cf. Figure 1). Red circles represent total retreat over the period between our earliest (BAARE) and latest (Landsat) data, and are sized proportionally according to the magnitude of retreat. See Figures 4 and 5 for higher temporal resolution changes derived from our imagery.

and 30th August 2004. The data was accessed from the Bolin Centre for Climate Research at Stockholm University (https://oden.geo.su.se/bathy/) with the assistance of A. Bjørk (personal communication, January 2021).

## 3 Results

### 285 3.1 Glacier frontal position

Figure 3 shows the overall net glacier terminus change that has taken place over the 89 year period of investigation. This figure gives unique insights into glaciological changes. This is unique to our study because this area is relatively little studied, and also unique because here we extend the record of change back beyond the era of imagery available from the satellite record alone. This gives important insights into glacier extent and change in the pre-satellite era. Figure 3 shows that over this time period 290 all glaciers in our study area experienced overall retreat. Significantly, those glaciers in the southern study region retreated by up to 3 km.

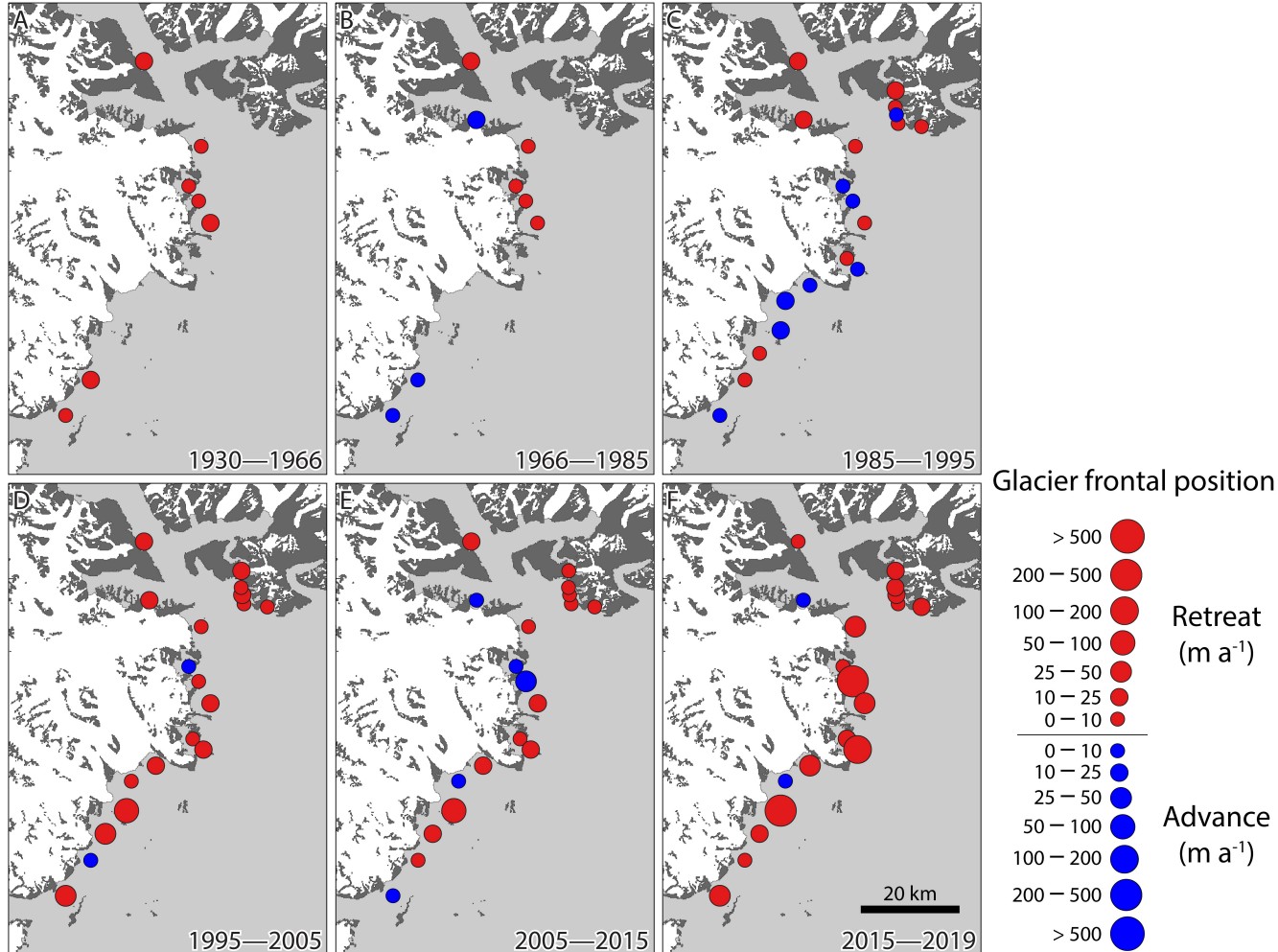

**Figure 4.** Graphical representation of annual rate of change in frontal position of glaciers in the northern region. Parts A to F show annual rates of change over the different time-steps that we have been able to generate from our various image sources. (A) BAARE mission (1930) to CORONA mission (1966). (B) CORONA mission (1966) to Landsat 5 (1985). (c) Landsat 5 (1985) to Landsat 5 (1995). (D) Landsat 5 (1995) to Landsat 7 (2005). (E) Landsat 7 (2005) to Landsat 8 (2015). (F) Landsat 8 (2015) to Landsat 8 (2019). Red circles represent retreat and green circles represent advance. Circles are sized proportionally to the magnitude of change. Most striking is the increased rate of glacier retreat in the most recent period (2015-2019).

In the northern study region, all glaciers again showed retreat. However, there was more variation with some glaciers in this relatively small area experiencing total retreats of several kilometres. Others showed much smaller amounts of retreat. Figures 4 and 5 break these frontal changes down into the individual time-steps available to us from our suite of imagery. These data are also summarised in Table 2. Figure 4 shows changes over each time step in the northern region (cf. Figure 1). In the earliest period (1930-1966) all glaciers appear to be retreating. By contrast, in subsequent time periods some glaciers appear to show

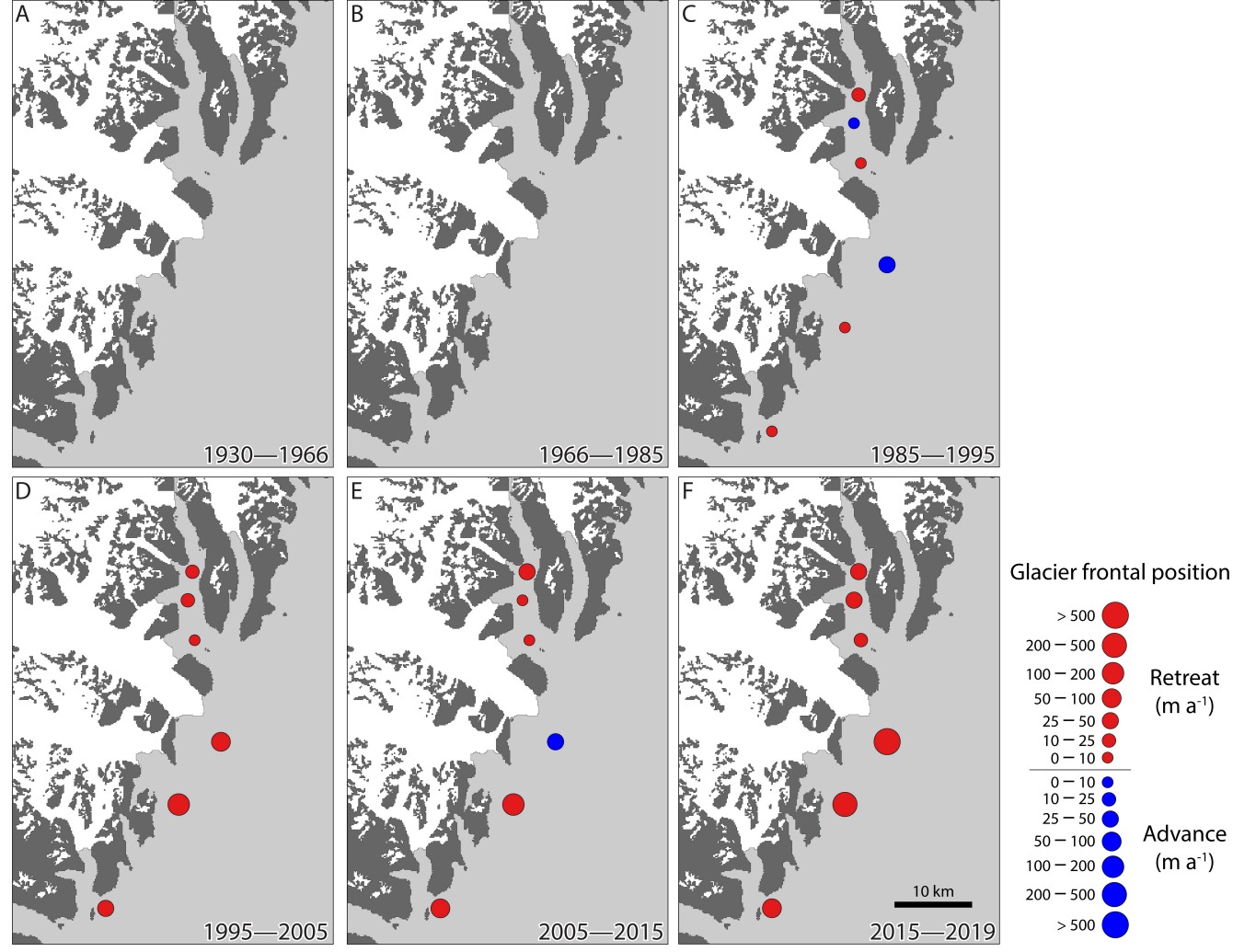

**Figure 5.** Graphical representation of annual rate of change in frontal position of glaciers in the southern region. Parts A to F show annual rates of change over the different time-steps that we have been able to generate from our various image sources. (A) BAARE mission (1930) to CORONA mission (1966). (B) CORONA mission (1966) to Landsat 5 (1985). (c) Landsat 5 (1985) to Landsat 5 (1995). (D) Landsat 5 (1995) to Landsat 7 (2005). (E) Landsat 7 (2005) to Landsat 8 (2015). (F) Landsat 8 (2015) to Landsat 8 (2019). Red circles represent retreat and green circles represent advance. Circles are sized proportionally to the magnitude of change. Note the lack of measurements in the first two periods. This arises because we do not have data for the 1960s for this region. As with the northern region, larger magnitude glacier retreat rates dominate in the most recent period (2015-2019).

small amounts of advance. Most noticeably, in the period 1985-1995 substantial advance took place for a significant number of the outlets. After 1995, although some glaciers continued to show advance of their termini, retreat dominated again and at an elevated rate as compared to previous time periods. In the most recent period (2015-2019), the vast majority of glaciers showed

| Glacier | 1930-1966 | 1966-1985 | 1985-1995 | 1995-2005 | 2005-2015 | 2015-2019 |
|---|---|---|---|---|---|---|
| 1 | | -13.3 | -0.2 | -6.3 | -5.9 | -13.8 |
| 2 | | -2.1 | -1.1 | -6.2 | -3.3 | -8.2 |
| 3 | | -4.1 | 2.7 | -11.1 | -9.5 | -10.5 |
| 4 | | -10.4 | -4.0 | -5.3 | -6.4 | -17.5 |
| 5 | | -4.5 | -11.5 | -10.9 | -5.0 | -14.0 |
| 6 | -15.6 | -13.8 | -13.5 | -22.0 | -21.2 | -6.8 |
| 7 | | | -10.5 | -19.9 | 9.0 | 0.2 |
| 8 | -9.2 | -7.2 | -0.6 | -9.0 | 8.3 | -32.3 |
| 9 | -1.9 | -1.4 | 2.3 | 0.4 | 1.2 | -8.8 |
| 10 | -2.3 | -4.0 | 3.3 | -5.3 | -29.9 | -298.8 |
| 11 | -18.3 | -12.8 | -1.6 | -11.1 | -14.4 | -41.8 |
| 12 | | | -3.4 | -2.0 | -0.6 | -21.0 |
| 13 | | | 4.2 | -12.4 | -13.1 | -115.8 |
| 14 | | | 13.8 | -13.4 | -8.0 | -14.5 |
| 15 | | -4.8 | 20.6 | -65.6 | -60.4 | -406.8 |
| 16 | | -21.2 | -2.2 | -33.5 | -13.6 | -21.5 |
| 17 | -14.5 | -9.3 | -4.2 | 9.4 | -4.8 | -2.5 |
| 18 | -6.3 | -3.0 | 6.2 | -38.7 | 9.6 | -48.8 |
| 19 | | | -21.2 | -22.4 | -33.4 | -37.0 |
| 20 | | | 9.4 | -21.1 | -3.5 | -34.8 |
| 21 | | | -2.2 | -5.5 | -3.1 | -14.25 |
| 22 | | 13.3 | 42.0 | -52.9 | 35.7 | -979.5 |
| 23 | | -23.1 | -6.5 | -171.7 | -120.2 | -318.8 |
| 24 | | -22.0 | -6.1 | -38.8 | -73.1 | 74.8 |

**Table 2.** Raw advance (positive)/retreat (negative) for all timesteps for all individual glaciers as numbered in Figure 1. All quantities are in metres per year. Missing values indicate that imagery of the glacier is not present in one of the two datasets used for that timestep. Estimated errors for glacier change for each time step is: 1930-1966: $\pm\,0.7\,\mathrm{m\,a^{-1}}$, 1966-1985: $\pm\,1.6\,\mathrm{m\,a^{-1}}$; 1985-1995: $\pm\,3.9\,\mathrm{m\,a^{-1}}$; 1995-2005: $\pm\,3.7\,\mathrm{m\,a^{-1}}$; 2005-2015: $\pm\,2.7\,\mathrm{m\,a^{-1}}$; 2015-2019: $\pm\,6.8\,\mathrm{m\,a^{-1}}$

retreat and at an accelerated rate as compared to previous time periods. Just two outlets (Figure 4) showed small amounts of advance.

In the southern region (Figure 5), there are fewer glaciers to consider. Indeed we do not have any data for the period 1930-1985. It is also important to note that we do not have data for the 1960s for this region and so parts A and B of Figure 5 are blank. As with the northern region, the 1985-1995 timestep was one in which advance of some glacier termini also took place,

while retreat was experienced by others. Moving beyond 1995, as with the northern region the retreat of most glaciers resumed at an increased rate in the most recent period (2015-2019).

In Figure 6, we divide the glaciers in our study regions according to type, and use box plots to visualise the range of variation in response of different glaciers. These plots reveal that there is substantial variability between individual glaciers, which is not unexpected due to the complexity of individual glacier response. We can identify a general trend from retreat during the early parts of our study period through to slight advance in the middle part of our study and then increasing rates of retreat in more recent decades. The marine terminating glaciers appear to be more dynamic, showing greater diversity of change between individual glaciers than land-terminating glaciers. However, both types of glacier show the same overall pattern, as described above. Taking ice-sheet outlets as a whole, we see that period of 1960s-1990s is one characterised by advance. Both before this period and since, retreat has been more dominant. In particular, more retreat is apparent since the turn of the 21st century. Local glaciers meanwhile, have displayed retreat throughout (with possible equilibrium in the 1990s) and perhaps a slight trend towards increasing rate of retreat in more recent years. Notably, land-terminating glaciers and local GICs that are peripheral to the GrIS (which also tend to terminate on land) show much less variability (i.e. shorter whiskers in the box plots). This may indicate a more consistent response to climatic drivers of change.

It is also worth noting that amongst the wide variability in behaviour, some glaciers demonstrated marked stability throughout the period of our study. The precise details pertaining to this will be dealt with in depth in the discussions.

### 3.2 Annual air temperatures

We see that temperatures in the early years of the 20th century are below the baseline originally defined by Box et al. (2009) but then rise well above it by the time of our first model in 1930 (Figure 7a). This figure shows annual air temperatures from the Tasilaaq meteorological station, provided by the Danish Meteorological Institute (https://www.dmi.dk/publikationer/). Data collection commenced at this location in 1895, and here we plot data over the course of the 20th and 21st century. White circles represent mean annual maximum temperatures while black circles represent mean annual minimum temperatures. Upper (maximum temperatures) and lower (minimum temperatures) red lines represent 10 year rolling means, and both of these track each other in terms of trajectory. Meanwhile, the black horizontal lines through each the different plots represent the minimum and maximum baselines (based on the mean values over the 1951-1980 period (Box et al., 2009)). The blue line is a calculation of the positive degree days over this period. Finally, triangles indicate timing of our glacier front observations from either aerial imagery (purple) or satellite imagery (green).

Between 1930 and our second model in 1965, temperatures remain above their baselines, but then dip below it again, remaining lower until the turn of the 21st century. After this time, temperatures remain above the baseline, and this is particularly pronounced for minimum mean temperatures. Minimum mean temperatures show greater variability around the running mean than maximum temperatures, and also show a pronounced trajectory of ongoing increasing values from the present into the future. Positive degree days shows similar general trends to that shown in the minimum and maximum temperature data. However, this is less obvious due to substantial variability from year to year.

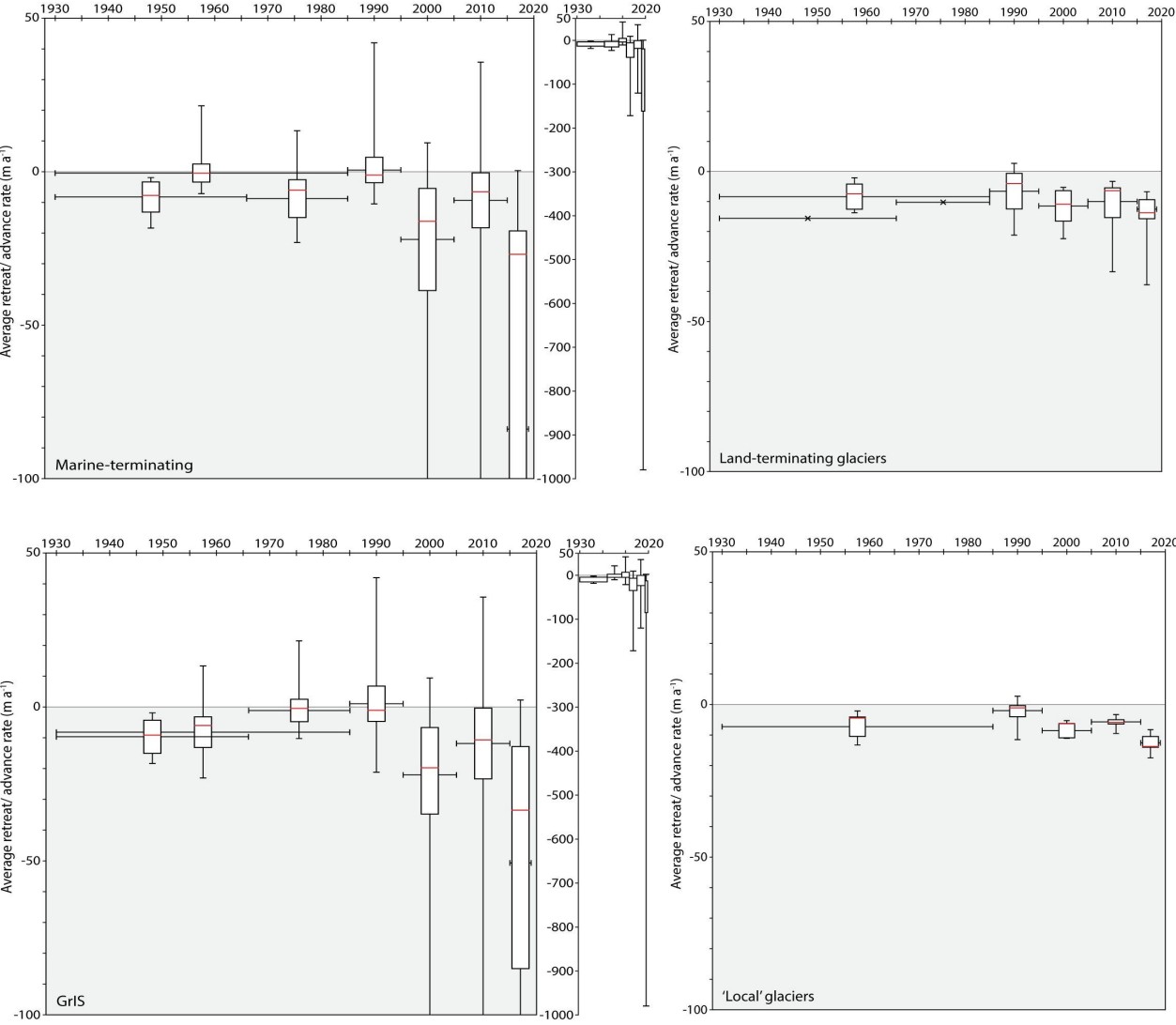

**Figure 6.** Box plots showing magnitude of change (i.e. advance or retreat) over the different time-steps. Vertical bars represent the range of measurements of change across all our glaciers, whilst horizontal bars represent the time period covered by a particular box. The box itself shows upper and lower quartiles with the median change in red and the whiskers show the maximum and minimum values. Here, we divide all glaciers under investigation across both regions (i.e. north and south) by type. We differentiate between marine-terminating glaciers (top-left) and land-terminating glaciers (top-right) of the Greenland Ice Sheet; and then also make a distinction between the behaviour of all outlets of the GrIS (bottom-left) and GICs that are peripheral to the GrIS (bottom-right). In addition, the narrow central plots are duplicates of the plots associated with marine-terminating glaciers (top) and all GrIS outlets (bottom) but with enlarged y-axes to show the full extent of error bars.

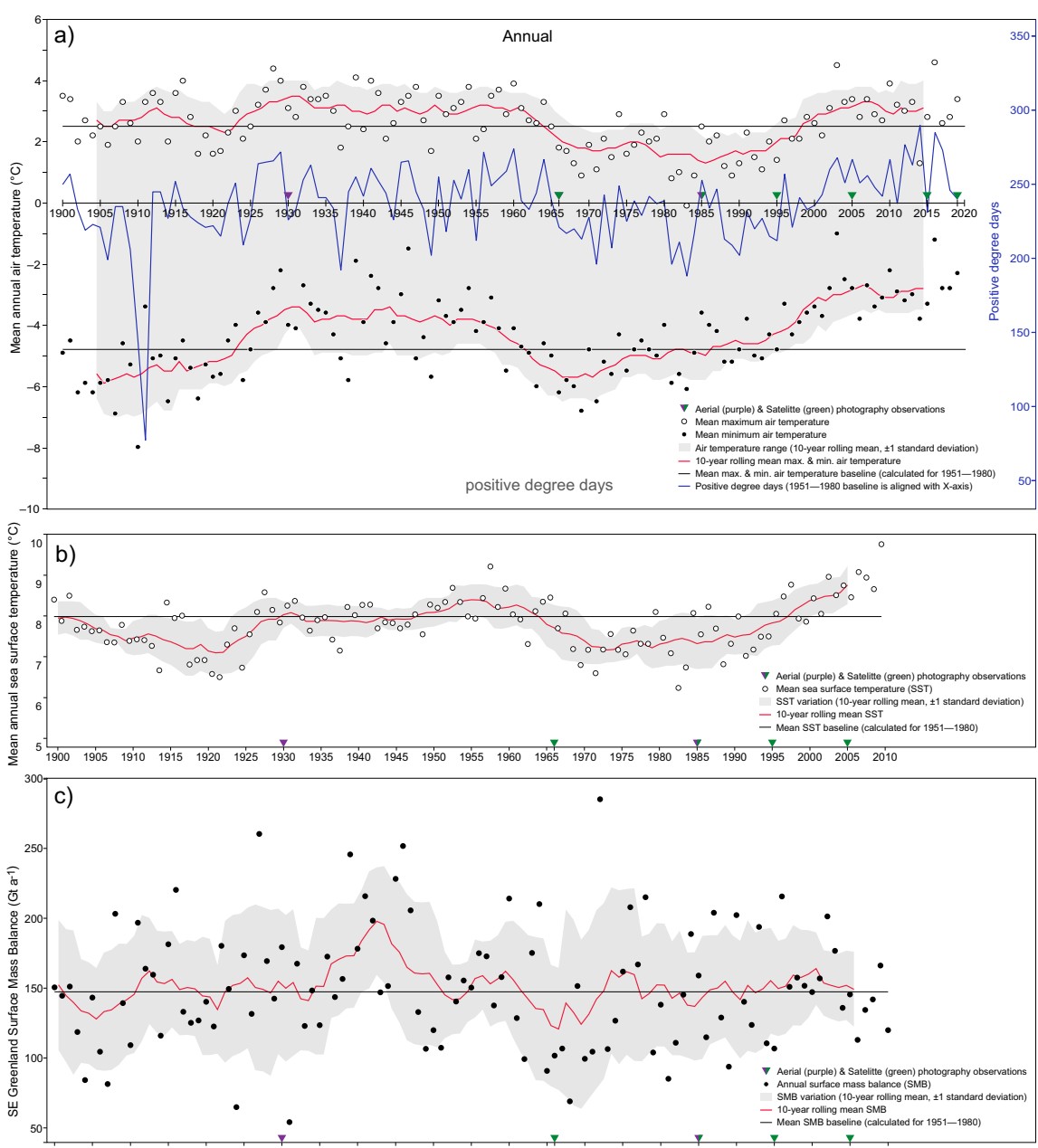

**Figure 7.** a) Annual air temperatures from the Tasilaaq meteorological station (data provided by the Danish Meteorological Institute; https://www.dmi.dk/publikationer/); b) Mean annual sea surface temperatures (SST) determined from the Hadley Centre Sea Ice and Sea Surface Temperature data set, taken from the Met Office Marine Data Bank (MDB; https://www.metoffice.gov.uk/hadobs/hadisst/index.html); c) Surface mass balance (SMB) in South East Greenland for Mittivakkat Gletscher (Mernild et al., 2011). More detailed explanation of each part of this figure is provided in the text.

### 3.3 Sea surface temperatures

Mean annual sea surface temperatures (SST) used in this study were determined from the Hadley Centre Sea Ice and Sea Surface Temperature data set (which is itself taken from the Met Office Marine Data Bank (MDB)). Detailed information on how this data set was created can be found in //www.metoffice.gov.uk/hadobs/hadisst/index.html and Rayner et al. (2003). In Figure 7b), white circles represent measurements of SST while the red line is a 10 year rolling mean. We compare these values with a baseline (black horizontal line) calculated as the mean SST of the 1951-1980 period (Box et al., 2009). Triangles indicate timing of our glacier front observations from either aerial imagery (purple) or satellite imagery (green).

We are able to explore SST over a period of more than one hundred years. During this period it fluctuates around the baseline (calculated as the mean of the SST over the period 1951-1980; Figure 7b)). In the first time-step (1930-1966) lower than average SST was initially apparent. This is followed by a rise above the mean later on. Immediately prior to this time-step, SSTs were below the mean. In the second time-step (1965-1985), SSTs dropped below the baseline. After 1985, SSTs start to increase and continued to do so, reaching values higher than at any preceding time. SSTs very closely track changes in annual air temperatures (Figure 7a), showing the same broadscale variation.

### 3.4 Surface mass balance

Surface mass balance (SMB) in South East Greenland (Wake et al., 2009; Box and Colgan, 2013) can be seen in 7c). Black circles represent measurements of SMB while the red line is a 10 year rolling mean. We compare these values with a baseline (black horizontal line) calculated as the mean SMB of the 1951-1980 period (Box et al., 2009). Triangles indicate timing of our glacier front observations from either aerial imagery (purple) or satellite imagery (green). There is a significant amount of variability in SMB over the 20th century, reflecting the importance of the 10-year running mean in order to discern longer term patterns in behaviour (Figure 7c)). It is important to note that over the entire 20th century, SMB was positive in this region of SE Greenland. Prior to our first model (1930), SMB fluctuated around the baseline SMB. Almost coincident with our first model (1930), SMB became increasingly positive. In the years following this (our first time-step of 1930 to 1966), SMB remained high before dropping down to the baseline by the end of the period. Low SMB dominated for a few years in the latter years of the 1960s and early years of the 1970s, before a sustained period of substantial variability around the baseline throughout the rest of our study period. Although only a few years of data are available, there is a slight negative trend in SMB since the turn of the 21st century.

### 3.5 Summarising frontal advance/retreat, SMB, SST and annual air temperature changes

Our investigation of SST, annual air temperatures and indeed SMB changes over both the 20th century and the early part of the 21st century is in an effort to explore likely drivers for the changes in ice front positions that we observe in the archival imagery. A glacier's terminus position is controlled by the balance between a) the amount of ice being added to the parent ice mass, b) the flow of this ice towards the terminus, and c) losses at the terminus induced by melt (either atmospheric or marine) as well

as potential iceberg calving. We are interested in investigating patterns between these controlling environmental variables. We also make links between these variables and the behaviour of the ice fronts explored in both our northern and southern regions.

Both air temperature and SST show the same broad trends, dividing the 20th and early part of the 21st century that is covered by our investigation into:

(i) an early period (up until ~1930) of cooler than baseline temperatures;

(ii) a sustained period of warmer than baseline temperatures (up to ~1965);

(iii) a shorter period of cooler temperatures (up to ~1990/1995);

(iv) a period of warming that continues up to the present day but which shows some flattening in recent years.

Of note, however, is that the SST fluctuation is much more subdued than that in the air temperature. This is particularly the case in period (ii), when SST is only very marginally above the baseline. There is also a lag, such that SST variations are not only subdued but also lag several years behind air temperatures. This is, of course, not surprising, since ocean temperatures rise as a consequence of atmospheric warming or cooling.

Variations in SMB do not track changes in air temperature or SST in a simple or direct way, but clear trends are apparent. Period (i) is one of fluctuating SMB around the baseline. Period (ii) is one in which SMB becomes increasingly positive before declining again. This is followed by continuing declining SMB and then further fluctuations around the baseline in period (iii) and (iv). Superimposed on this is a higher frequency variability in SMB. A simple assumption that SMB responds directly to time-integrated air and/or SST changes is therefore not apparent, nor would it be expected. The associated complexity is a consequence of how the controls on energy inputs into large ice masses vary on a range of timescales, and also the temporal lag between these inputs and an ice mass responding. It is also a consequence of other controls on ice mass response, such as changing oceanic circulation patterns and geomorphological controls. Explanations for the complexity of the response are considered in detail in the discussions. It is interesting to observe that there are clear and broadscale patterns in the SMB response that could be attributed to variations in air temperature and SST.

We also observe these external forcings playing out in the changing extent of the outlets, but with a degree of complexity possibly reflecting a lag in the response of the ice masses. For instance, the cooler period (i) does not immediately manifest itself as ice front advance during this same period. Rather it is some years later (most notably Figures 4b and c) where we see glacier advance. This is also reflected in the positive SMB in period (ii) which clearly manifests itself as advance of many outlets. Sustained and widespread retreat at a growing rate commences just as period (iv) begins, continuing up to the present day. There is thus general and broad scale manifestation of these changing parameters in both SMB and glacier frontal response. However, not all outlets respond in the same way, and so these too are considered in the discussions.

## 4    Discussion

### 4.1    Outlet response to regional climatic trends

Our work with archival imagery has enabled us to extend the record of glacier frontal change beyond the limits of the satellite record. We have also been able to do this in a region of Greenland that is relatively poorly studied. We have shown that glacier frontal positions varied over this time period, alongside limited measurements of surface mass balance. These varying glacier extents occur in response to changes in both air temperatures and SSTs, which fluctuate between cool/warm/cool/warm conditions (around our baseline). This suggests that the underlying drivers of these changes are air and ocean temperatures.

In general, in our data we see that the overall (regional) trends in glacier change (as observed in the box plots of Figure 6) do track the prevailing climatic forcing. Greater rates of retreat take place during the warmer period (approximately 1925-1964), with a more subtle slowing of this response during the cooler periods (approximately 1905-1925 and 1964-1996), and a faster retreat/collapse in the contemporary period (approximately 1996 onwards) (Hanna et al., 2012; Van den Broeke et al., 2016). Hanna et al. (2021) suggest that Greenlandic air temperature trends are generally flat since 2001. Taking this period in

isolation, although there is some clear variability, our data also shows that the overall trend is flat or at least subdued. However, considering the contemporary period as a whole, we believe that temperature trends do show an overall increasing trend. This is particularly so in the record of minimum air temperature, which may be significant when considering the role of elevated minimum temperatures on the net amount of melt that takes place. In the contemporary period, we also see warmer seas, as well as a larger increase in positive degree days. There is considerable variability from year to year in the positive degree days

during this period, which perhaps reflects the compensating short term warming and cooling events referred to by Hanna et al. (2021).

Our observations in relation to SST are in close keeping with recent work by Wood et al. (2021). This shows that the speed up and mass loss of Greenlandic glaciers since the mid-1990s has been as a consequence of warm Atlantic ocean waters intruding into fjords. They conclude that nearly one third of their sample of 226 marine terminating glaciers owe nearly half of their

mass loss to these warming waters. We hypothesise that warming ocean waters may well play an important role in the mass loss we observe. It is, however, important to note that the focus of Wood et al. (2021) is on subsurface water temperatures that occurred as a result of the spreading of ocean heat caused by changes in the North Atlantic Oscillation (NAO). We do not have data that enables us to explore subsurface temperatures in this way.

However, glacier frontal response to these climatic drivers is more complex and time-lagged. Of course not all glaciers

respond in the same way, with the same magnitude or at the same rate. This indicates that there are additional controls too. Figure 6 demonstrates this significant heterogeneity. Here we have subdivided the glaciers in our study area according to type. We see that glaciers of a different type respond differently to external drivers. The marine terminating glaciers in our study region are a) more dynamic, b) show more retreat and c) show more varied behaviour than land terminating glaciers. Such behaviour is well-documented (Moon and Joughin, 2008; Murray et al., 2015), and highlights that the oceans (currents, tides

and bathymetry) and SST changes (as well as subsurface temperature changes) have a vital role in the stability of these ice masses.

Such complexity of response, and variability amongst marine terminating glaciers is also an observation reported recently by Wood et al. (2021), and which is discussed in more detail below. For local glaciers and land terminating glaciers, we observe that whilst in earlier periods, changes in these glaciers were relatively small, larger changes have become more apparent recently. We propose that this 'switch' could be representative of SMB becoming an increasingly important driver of change in recent years, as has been documented elsewhere (cf. Wood et al. (2021)).

## 4.2 Local heterogeneity in glacier response

As well as differing behaviour of different types of ice mass, we also observe significant local heterogeneity in glacier response - i.e. glaciers that are close neighbours and are of the same type can also show very different behaviour. This is an important observation, since neighbouring glaciers are subject to the same external drivers. Therefore differing responses implies there are significant additional processes in operation. Such observations suggest that glacier response is defined not only by climatic variables (e.g. air temperature, SST) but also by a) ice velocity (and changes in this over time; King et al. (2020)), b) ocean circulation at a calving front (Wood et al., 2018), c) underlying topography (i.e. bed elevation beneath an ice mass) and bathymetry (Catania et al., 2018), d) the presence, concentration and role of sea ice (Carr et al., 2013), and e) ice thickness (Bahr et al., 1998). Of course, the role of SST, ocean circulation, bathymetry and sea ice are only relevant controls with respect to marine terminating glaciers.

Figures 8 and 9 focus on the two sub-areas of our northern region (see Figure 1), and in particular two sets of glaciers (Glaciers 7, 8, 9 and 10 in Figure 8, and Glaciers 14 and 15 in Figure 9) which show significantly different behaviour despite being proximally located. In Figure 8, it is apparent that many glaciers in the region show considerable consistency of behaviour, with little frontal change over the study period. Of particular note is the stability of many of the outlets, such that over the ~90 year period of investigation, only relatively small amounts of retreat have manifested. This is despite it being recognised that there have been significant mass losses to the oceans in recent decades. The most significant mass loss has been since 1998. Since this date there has been annual mass loss from Greenland in every year (Kjeldsen et al., 2015; Mouginot et al., 2019). The recent investigation of Wood et al. (2021) explores the role of ocean forcing in Greenlandic glacier retreat. The study attempts to categorise glaciers according to their geometry and water-depth. In the area covered by our study, the vast majority of glaciers (in fact all but one) are described by Wood et al. (2021) as 'noncategorized,' which means that the bathymetry and water properties are unknown. Our long-term investigation of these glaciers and the observation of their apparent stability, suggests these glaciers may sit in relatively shallow water on shallow ridges. This prevents the intrusion of warm deep water which would further enhance mass loss (Wood et al., 2021). Although their bathymetry is currently entirely unknown (Wood et al., 2021), it is possible that in the future these glaciers may pass a tipping point when they retreat off their pinning ridge into deeper waters. This would see a switch from their current status of having little frontal change, to a phase with much more rapid retreat. At present this is very much speculative, but ongoing monitoring of these outlets is therefore of great importance.

In addition to these previously discussed outlets in which frontal change is minimal, there are three outlets that in particular show 2-3 times more retreat than these. Two of these are part of Glacier 8 (see Figure 8) which has several outlets. The two

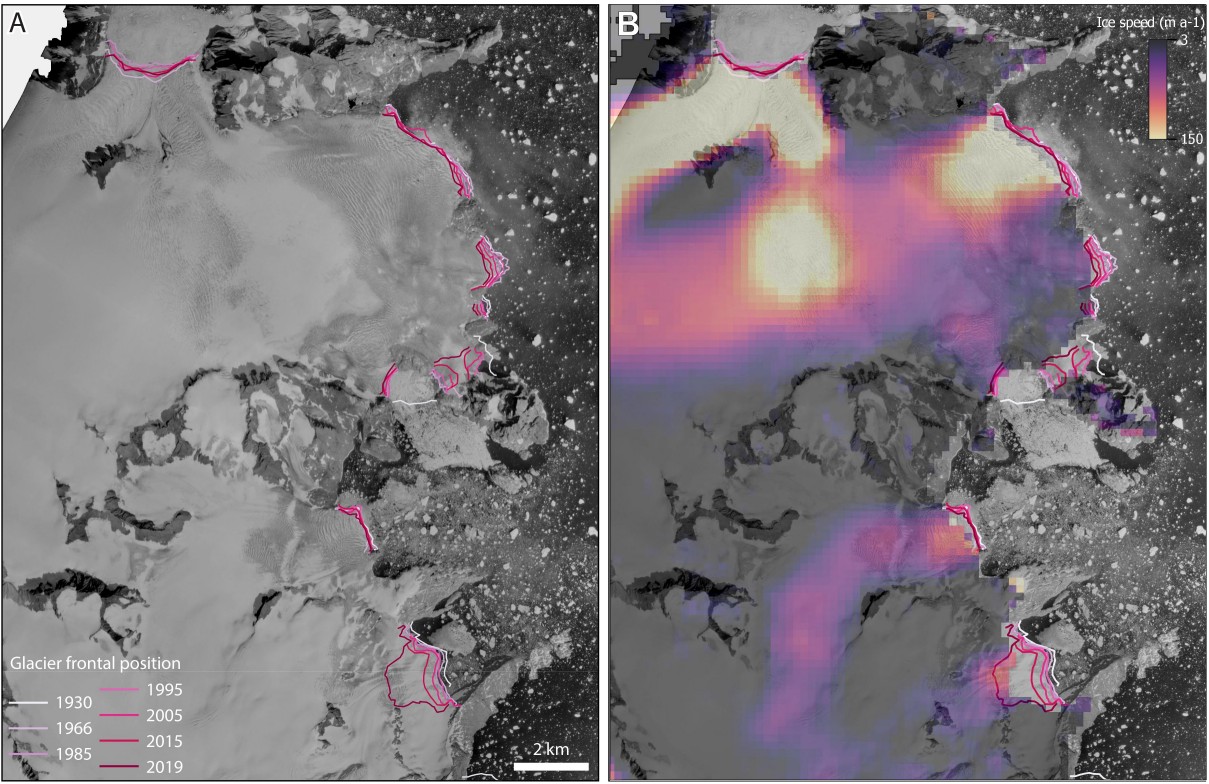

**Figure 8.** Position of glacier margins in part of the upper northern region (see Figure 1). (A) Frontal positions are shown for the years 1930, 1966, 1985, 1995, 2005, 2015 and 2019. for glaciers 8, 9 and 10 as shown in Figure 1A. (B) Frontal positions are again shown, but with colouration indicating ice velocity as well. Ice velocities derived from Joughin et al. (2010) and the background is the 80s mosaic.

southernmost ones showed rapid and large-scale retreat between 1930 and 1966, but then displayed very little change over the years since then. In contrast, Glacier 10 showed relatively modest retreat from 1930 to 2015, but then large-scale and rapid retreat in the four years to 2019 (Figure 8). Figure 9 shows similar behaviour. Here, Glacier 14 appears to be very stable, with minimal fluctuation around its terminus over the duration of the study period, albeit with an overall trend towards modest

retreat. However, there is some complexity within this glacier alone. This is due to the fact that the eastern side of this very wide marine-terminating outlet shows more consistent and substantial retreat. Glacier 15 shows much more retreat with several large retreat 'steps', but with the most significant retreat step being between 2015 and 2019.

Wood et al. (2021) similarly reported that many of Greenland's marine-terminating glaciers have sped-up and lost mass as a consequence of warming ocean waters, but that there are some glaciers that have exhibited small or no retreat. The explanation

presented by them for this minimal retreat is that this is a result of water being shallow or outlets resting on shallow ridges. It may well be that this also helps to explain the diversity of behaviour we identify. Many of our study glaciers show little retreat over the 90 year study period and although we do not have bathymetry data, we propose that these understudied glaciers also sit on ridges and/or in shallow water. Where our glaciers have shown periods of more significant retreat for some part of the 90

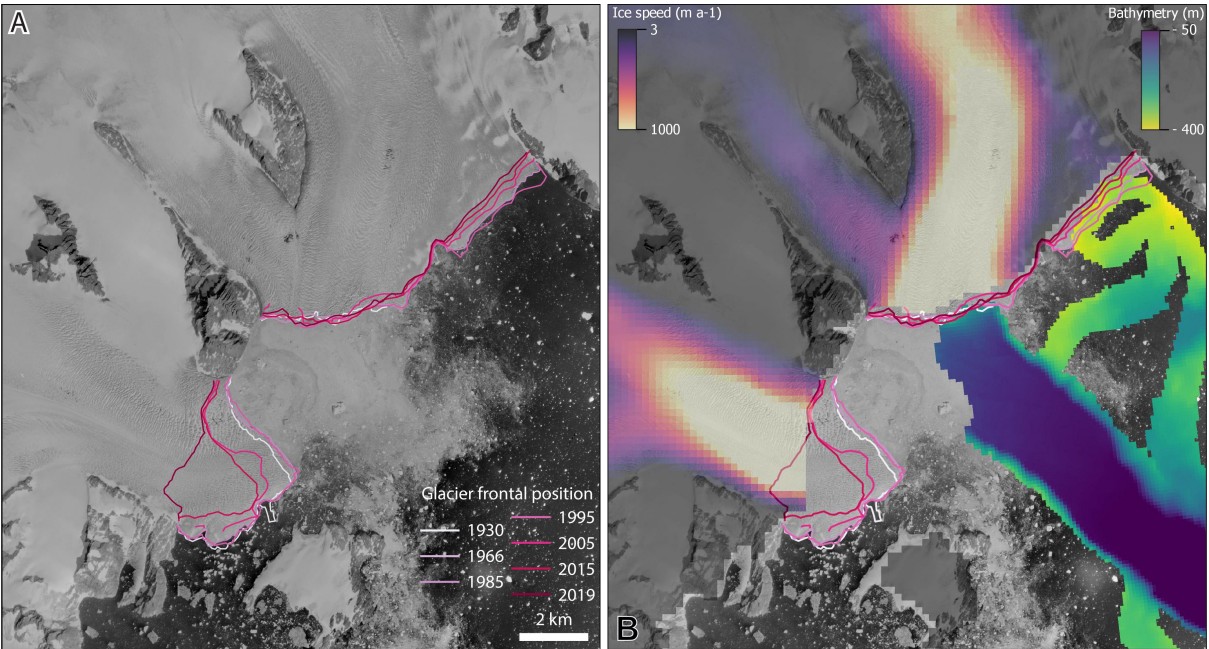

**Figure 9.** Position of glacier margins in part of the lower northern region (see Figure 1). (A) Frontal positions are shown for the years 1930, 1966, 1985, 1995, 2005, 2015 and 2019. for glaciers 14 and 15 as shown in Figure 1A. (B) Frontal positions are again shown, but with colouration indicating ice velocity as well. Ice velocities derived from Joughin et al. (2010) and the background is the 80s mosaic.

year investigation, we propose that these periods of change indicate when glaciers move off pinning ridges into deeper water,
even though they then may subsequently become grounded again and thus their retreat slows.

With regards to the differing behaviour of the two parts of the front of Glacier 14 (see Figure 9), we are fortunate to have bathymetry data (A. Bjørk, personal communication, January 2021) for the region directly abutting the ice front (Figure 9B). This reveals starkly different topography, with that in front of the more stable region being significantly deeper than that in front of the more changeable region. However, on closer inspection of the bathymetry data it is apparent that directly in front
of the western part of this glacier, there is a subtle shallowing of the bed. This could suggest the presence of a ridge which pins the glacier and thus explains why it appears to have a stable front. We do see higher ice velocities here (see Figure 9B) and so it is also possible that the apparent relative stability of of the western outlet arises because comparatively high calving rates are offset by higher ice velocities delivering ice more rapidly to the ice front. Conversely, the eastern outlet lies in shallower water but retreat is nevertheless more substantial. Ice velocities are lower here and so calving and/or melting is not countered by ice
flow from inland (i.e. lower velocities than in the west (Figure 9B)).

Finally, Glacier 15 shows significant frontal retreat and high surface velocities. Following the thinking described above, we propose that this suggests that despite the delivery of large amounts of ice to the calving front from inland, significant retreat is still occurring, and so this glacier may be losing the greatest amount of mass overall.

Such diverse observations highlight how even dividing the ice sheet up into regions masks the complexity that is inherent in individual glacier behaviour. Even glaciers that exist adjacent to each other can show markedly differing patterns of retreat. Significant variability in the behaviour of Greenlandic outlet glaciers has been identified previously (McFadden et al., 2011; Moo; Csatho et al., 2014; Porter et al., 2018), whereby variations in rates of frontal retreat, surface thinning and velocity may be apparent even when a region as a whole is losing mass. An individual glacier has a unique mixture of processes that might control its rate of retreat and thus it is oversimplistic just to state that mass balance or dynamic changes dominate in a region. In particular, we propose that our long-term study of multiple glaciers suggests a very important role for subglacial and submarine topography and in particular the importance of shallow ridges that dictate the retreat rates of marine-terminating glaciers. Porter et al. (2018) used a statistical approach to explore the spatial correlation in the behaviour of adjacent Greenlandic glaciers and showed that local controls are more important than regional influences. They also found that there was a good correlation between rate of thinning and ocean heat content, and also that glaciers grounded in deeper water were more sensitive to oceanic controls on mass loss. Similar to our findings, they also showed that taking account of the presence of shallow sills was important, and further called for an improved understanding of bathymetry. Catania et al. (2018) also revealed how fjord geometry is an important control on how glaciers respond to climate. However, their work was focussed in West Greenland and explored changes over the past 30 years. The novelty of our work is not only the exploration of previously unstudied, smaller outlets in East Greenland, but also that we are able to identify such processes taking place over a timescale that is three times as long, thanks to the data we are able to extract from archival imagery. This greater length means that we can see that there was an earlier and a later warmer period, and that the glaciers responded differently in each, such that much more retreat took place in the later period. If topography is indeed the control on this, then it demonstrates the significance and ability to be a major moderator of climate-driven changes.

## 5 Conclusions

Our investigation has shown the potential of archival imagery that was not originally (and thus not optimally) collected for the purpose of photogrammetric investigations of glacier change. It is thus an important demonstration of the powerful quantitative data that resides in such imagery. This archival imagery has enabled us to extend the record of change of a number of little studied glaciers that reside in the central-eastern part of Greenland (Mouginot et al., 2019), back by several decades beyond the beginnings of the satellite record. Being able to do this is of great benefit since a longer time series of glacier change enables a better understanding of how ice masses have responded to climate to be developed (Dyurgerov and Meier, 2000).

Our focus here has been on a series of outlet glaciers from the Greenland Ice Sheet, and an investigation of how these have varied alongside a number of other controlling environmental parameters for which we also have long-term records. Our study covers ~90 years and is the first such dedicated study in this region and over this duration. It deals with changes of a number of previously poorly studied glaciers that have perhaps been largely overlooked. One of our key findings is that climate forcing exhibits strong controls on glaciers in the region generally, and that there is a very close link between air temperatures and

SSTs. Arguably, SSTs are more important as we see larger scale significant retreat of outlets terminating in water as the oceans have warmed.

However, our study region contains a number of different types of glacier. We observe that it is the marine terminating glaciers that show the greatest mass loss, particularly in the more recent period. Aside from our observation of the importance of climatic forcing, we also highlight significant local variations and the potential importance of non-climate-related factors. Above all, one of our primary conclusions is that there is enormous variability in how glaciers respond to the climatic and non-climatic drivers. In particular, we propose that the great variability in the retreat of marine terminating glaciers (both in terms of the magnitude and timing of retreat) may be controlled by the presence or lack of shallow ridges which act to pin glaciers as they retreat. In our interpretation, we envisage an undulating submarine/subglacial topography which has meant that some glaciers have showed periods of much greater or lesser retreat, and some are apparently stable in their position. Such a situation, if accurate, would lend itself to the possibility of future periods of comparatively rapid retreat of glaciers that appear to be stable, and likewise future stabilisation of other glaciers that may currently (or in the past) have shown more significant retreat. Catania et al. (2018) also provided such insights for western Greenland and so we have greater confidence in our interpretation here. The novelty of our investigation is not only that we have shown such behaviour in a previously unstudied region of Eastern Greenland, but also that our use of archival imagery allows us to identify that such behaviour has been occurring over a longer time period than it has been previously able to show. This helps to demonstrate the rich insights that can be gained from the processing pipeline we demonstrate here. In the past, regional investigations across the Greenland Ice Sheet have been key (e.g. Mouginot et al. (2019); King et al. (2020)). This has been important for exploring broad scale regional behaviour and responses. However, our work here, in which we have focused on glacier-to-glacier heterogeneity, shows that within regions there is great complexity, with even adjacent glaciers behaving very differently. In our efforts to better understand the complexity of the response of the Greenland Ice Sheet to a warming climate, we propose that it is increasingly important to consider the variability between outlet glaciers because of the variation in responses that we have identified here. We also support, and further stress, the need for much improved knowledge of fjord geometry, as initially called for by Porter et al. (2018) because of its probable importance in controlling the heterogeneity in glacier behaviour. In addition, our work has also highlighted how difficult it is to analyse overall glacier response from investigations of frontal variations alone. An important future direction would be to focus on surface elevation change and also to explore the subglacial topography of these outlets to predict likely future 'jumping' periods of retreat, or indeed stabilisation.

*Code availability.* There is no specific code to be made available, but we will happily discuss our approach on request.

*Data availability.* Data will be available in the Pangea open access data repository

– Front changes basing on Landsat data: doi.org/10.1594/PANGAEA.941995

– Ortomosaics: doi.org/10.1594/PANGAEA.942134

Archival data sets are available thru there respected curators:

– The BAARE expedition dataset has been given to the Scott Polar Institute and can be obtained from there picture library web-page: https://www.spri.cam.ac.uk/picturelibrary/

– The CORONA can be found under: Declassified Satellite Imagery - 1 Digital Object Identifier (DOI) number: /10.5066/F78P5XZM and can be obtained from USGS (United States Geological Survey) via EarthExplorer.

– The 1980s areal images can be obtained from: The Danish Agency for Data Supply and Efficiency: https://eng.sdfe.dk/

*Author contributions.* Michael Cooper and Paulina Lewińska carried out the vast majority of the data processing, with additional guidance and contributions from William Smith, Edwin Hancock and David Rippin. David Rippin and Paulina Lewińska wrote most of the text, 565 with additional contributions from William Smith, Edwin Hancock and Michael Cooper. Julian Dowdeswell provided important additional insights and contributions.

*Competing interests.* There are no competing interests are present.

*Acknowledgements.* This work is funded by a Leverhulme Trust Research Grant entitled: 'Archival Polar Photography - Unearthing the Forgotten Record of Glacier Change' (reference: RPG-2017-346). Historical imagery was provided by the Picture Library at the Scott Polar 570 Research Institute, University of Cambridge, through Julian Dowdeswell and Lucy Martin. We acknowledge the Danish Meteorological Institute, the Bolin Centre for Climate Research at Stockholm University and the Met Office Marine Data Bank (MDB) as sources of other data-sets, and also gratefully acknowledge the assistance of A. Bjørk in gaining access to several datasets. The CORONA dataset was obtained via USGS EarthExplorer from the U.S. Geological Survey (Declassified Satellite Imagery - 1 Digital Object Identifier (DOI) number: /10.5066/F78P5XZM). Finally, we thank the editor, Etienne Berthier, and three anonymous reviewers for their insightful comments 575 that have helped to improve this manuscript.

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
