# Peer review of "Unravelling the long-term, locally-heterogenous response of Greenland glaciers observed in archival photography"

_The Cryosphere, 2021_

## Referee Comment (RC1)

**Review of tc-2021-256: Unravelling the long-term, locally-heterogenous response of Greenland glaciers observed in archival photography**

This manuscript presents orthorectified aerial and satellite imagery of the margin of the central-eastern portion of the Greenland Ice Sheet. Glacier terminus positions are digitized from the newly processed imagery, providing measurements of glacier extent in 1930 and 1966. These are combined with glacier terminus position measurements from satellite imagery from 1985, 1995, 2005, 2015, and 2019 to produce a ~90-year record. The glaciers are divided into categories of: marine-terminating, land-terminating, ice sheet outlets, and local glaciers. Their terminus position changes are interpreted in the context of changes in regional air temperature, positive degree days, sea surface temperature, and reconstructed surface mass balance. Finally, the heterogeneity between glaciers as well as within different portions of individual glaciers is discussed.

Processing the archival photography from the 1930/31 British Arctic Air Route Expedition and the CORONA satellite mission is a big undertaking. I'm very pleased to see this imagery being utilized to provide longer-term context for the recent changes that glaciers have undergone. Another strength of the paper is the focus on lesser-studied glaciers, including land-terminating ice sheet outlets and peripheral glaciers and ice caps, which shows clear differences between the behavior of these different types of glaciers. Finally, the exploration of heterogeneity between neighboring glaciers and heterogeneity within individual glaciers themselves (i.e., different branches or portions of the same glacier) is a strength.

There are, however, several improvements that can be made to strengthen the paper and I have noted these below in general and editorial comments. One overarching comment that I would like to emphasize is that the discussion needs improvement in order to better support the point that "full understanding of the behaviour and response of the Ice Sheet requires us to consider changes that are taking place at the scale of individual glaciers" (abstract). The paper does an excellent job of showing the heterogeneity among glaciers in the data, but the discussion is weak on interpreting this heterogeneity. I understand that, without observations at most of these glaciers (e.g., bathymetry, oceanographic data), it is not possible to draw conclusions about why some glaciers are stable while others retreat. This paper hypothesizes that bathymetry and subglacial topography is likely what is stabilizing some glaciers, but this has been shown in previous work (e.g., Catania et al., 2018). The discussion in the paper needs to be strengthened by showing what can be learned from focusing on the lesser-studied glaciers and by extending the record of glacier behavior back in time, beyond what has already been shown in past literature. In other words, the discussion can be improved by focusing on the novel aspects of the archival data. For example, one question that the authors can consider addressing is: if air and sea-surface temperatures exhibit similar anomalies from the long-term baseline in the 1930s and 1940s as they do in the 2000s and 2010s (Figs. 7 and 8), why do marine-terminating and GrIS outlets respond with so much more retreat in the latter period (Fig. 2)? This is a novel question that could be addressed with the archival data of southeast Greenland that could not have been addressed before.

I would just like to emphasize, again, that the archival photos processed for this study add crucial and novel measurements that help contextualize glacier changes that occurred over the last couple of decades. The processing of the data is thorough, and the results are presented clearly. I feel that, with some improvement to the discussion, this paper can provide an impactful contribution to our understanding of glacier change.

**General comments**

1. One of the conclusions of the study is that it is important to study glacier-to-glacier heterogeneity in order to get a full understanding of the behavior and response of the ice sheet. While I agree with this sentiment, I feel that this paper frames past work incorrectly. For example, lines 478-480 state that "there has been a temptation in the past to differentiate between regions." I disagree that past work that focused on regional aggregations of glacier behavior did so simply because it was tempting to do this. The goal of grouping regionally is to understand whether there are broad patterns to glacier behavior and their response to external forcing. And, while there have been previous studies that have grouped glaciers regionally, there are also studies that have shown significant heterogeneity between neighboring glaciers (e.g., Bjørk et al., 2012; Carr et al., 2017). I have two suggestions:
   a. Change the wording throughout the manuscript to more accurately state that the goals of the past regional studies was to investigate broad-scale patterns, while this present study focuses on glacier-to-glacier heterogeneity.
   b. Add to the introduction an overview of past studies that have looked at glacier-to-glacier heterogeneity, explain what they had found, and then go on to describe how this present study builds on that previous work.

2. An estimate of observation errors for SST and SMB should be added to the manuscript. Currently, Figures 7, 8, and 9 show the standard deviation of the data around the mean (gray shading) but this does not take into account errors in each individual measurement. I appreciate that it is difficult to assess the errors on these data but, at the very least, a rough estimate should be incorporated such that the gray shading includes both the scatter in the measurements and the errors on the measurements themselves.

3. I suggest combining Figures 7, 8, and 9 into one. This would make it easier to compare air temp, SST, and SMB.

4. Some of the text in the discussion section presents results without any interpretation and the manuscript would be clearer if this text were moved to the results section. I've tried to note this in my editorial comments below but I may have missed some. I suggest the authors go through the manuscript to move any text that does not include interpretation from the discussion section to the results section.

5. In general, more explanation is needed throughout the discussion to substantiate some of the interpretation. I've noted this in editorial comments below.

6. The interpretation of the behavior of Glacier 14 needs to be clarified. First, the text on lines 442-446 states that the bathymetry data in front of Glacier 14 shows that the stable portion of the terminus is located where the bed is deeper. Although the bed may be deeper in front of the terminus, the data doesn't show what the topography is just upstream of the terminus. There could potentially be a steep ridge there and the bathymetry data that's available does show a shallowing towards the terminus. Second, the text on lines 446-449 goes on state that the observed stability is in opposition to the fact that calving rates tend to be higher for termini in deeper water. This is slightly confusing because terminus stability arises from stable calving and melt rates. In other words, the calving rate can be high but as long as it doesn't increase (and melt rates don't increase), the terminus will not retreat (velocity also has to remain steady). So, it isn't the calving rate itself but changes in the calving rate that are important. Finally, this paragraph hypothesizes that the western part of this glacier is "subject to dynamically-driven mass loss" even though the terminus position is stable. This interpretation is not supported and I would like to see more explanation for why the authors think this is the case. The evidence presented is that the western portion has faster ice flow and is grounded in deeper water

(although I question this) but I don't see why this leads to the claim that this glacier is dynamically losing mass.

7. Similar to my previous point, the text on line 455 states that Glacier 15 is undergoing "surface thinning" and is "losing the greatest amount of mass" however this is not supported. Please add justification for these claims.

8. The data availability statement does not comply with the requirements of the journal. The journal data policy states that if the data is not made publicly accessible in a FAIR repository, an explanation needs to be provided for why this is the case (https://www.the-cryosphere.net/policies/data_policy.html). My strong recommendation is for the orthophotos from the BAARE and CORONA, as well as the derived terminus positions from all of the imagery (including those from Landsat), be placed in a FAIR repository, with a DOI obtained and referenced in the Data Availability section.

**Editorial comments**
[line 279] Change "Thss" to "This"

[lines 308-309] The sentence "Those outlets ... greater variability" repeats what has already been said earlier in the paragraph about marine-terminating glaciers. I suggest removing this sentence.

[lines 309-310] I suggest moving this sentence to the discussion because it does not directly describe results.

[lines 311-312] These sentences should be moved to the paragraph where Figures 2 and 3 are described.

[line 329] I suggest renaming this section to "Surface mass balance"

[line 365] I suggest changing "controllers" to "external forcings"

[lines 365-371] This paragraph should be moved to the discussion. It can be combined with the existing text in Section 4.1.

[lines 380-382] Please add the start and end years that define each of these periods discussed in this sentence

[lines 383-384] Please add some discussion of this differences in air temperature from this study and what was shown by Hanna et al. (2021). Is this just the difference between Greenland-wide air temperature versus the trend in the southeast? Or is there a difference between the air temperature reconstructions for the same region?

[line 385] Please explain why this increase is seen as important. Was there a threshold crossed? Is it the largest observed increase in XX years? I also suggest mentioning the positive degree days in the contemporary period and discuss their behavior.

[lines 391-392] More discussion is needed to explain why the interpretation is that warming waters are responsible for glacier retreat, rather than rising air temperatures.

[line 400] Add the word "detail" following "more"

[lines 402-404] This sentence, which states that SMB is a driver of marine-terminating glacier change in recent years, seems like it contradicts the previous paragraph, which states that ocean temperature has been responsible for the retreat of these glaciers since the 1990s. I suggest combining the previous paragraph with this sentence and clarifying the interpretation of whether one or both drivers are responsible for frontal changes during various periods.

[line 405] Remove the hyphen between "local" and "heterogeneity"

[lines 430-438] This paragraph presents results rather than interpretation. I suggest moving this text from the discussion to the results section.

[line 430] Please specify: "2-3 times more retreat" ... more than what?

[line 453] Should "down" be changed to "due"?

[lines 439-441] The first sentence needs to be rephrased because I think it incorrectly presents the conclusions of Wood et al. (2021). In fact, the second sentence of this paragraph contradicts the first. I suggest replacing the first two sentences with the following: "Although the behavior of many of Greenland's marine-terminating glaciers has been speed up and mass loss as a consequence of warming ocean waters (Wood et al., 2021), there are glaciers that have exhibited small or no retreat."

[lines 439-455] This paragraph hypothesizes the response of glaciers in terms of dynamic thinning and links that to glacier ice speed and calving. However, I'm not clearly seeing the link. For example, it is stated that the eastern portion of glacier 15 is sitting on a shallower bed and has slower ice speed and, therefore, the terminus retreat is due to calving and there is less dynamic thinning. Why would calving-driven retreat and slower ice speed necessarily imply that there is relatively less dynamic thinning? This link needs to be explained in greater detail.

[line 443] I believe that "Glacier 15" should be replaced with "Glacier 14" here

**References**
Bjørk, A. A., Kjær, K. H., Korsgaard, N. J., Khan, S. A., Kjeldsen, K. K., Andresen, C. S., et al. (2012). An aerial view of 80 years of climate-related glacier fluctuations in southeast Greenland. Nature Geoscience, 5(6), 427–432. http://doi.org/10.1038/ngeo1481

Carr, J. R., Stokes, C. R., & Vieli, A. (2017). Threefold increase in marine-terminating outlet glacier retreat rates across the Atlantic Arctic: 1992-2010. Annals of Glaciology, 58(74), 72–91. http://doi.org/10.1017/aog.2017.3

Catania, G. A., Stearns, L. A., Sutherland, D. A., Fried, M. J., Bartholomaus, T. C., Morlighem, M., et al. (2018). Geometric Controls on Tidewater Glacier Retreat in Central Western Greenland. Journal of Geophysical Research: Earth Surface, 29(1), 1–15. http://doi.org/10.1029/2017JF004499

---

## Author Response (AR1)

Dear Editor

Many thanks for overseeing the review of our paper: '*Unravelling the long-term, locally-heterogenous response of Greenland glaciers observed in archival photography*'. First of all, we would like to apologise for initially not responding to your own general comments. For some reason, we did not see these until your most recent correspondence asking us to make the corrections to our manuscript - prior to that, we were only away of your comments on the annotated manuscript. We do, however, now deal with these below. With regards to the reviewers' comments, we are very pleased to see the positive comments made by both reviewers on our work. We appreciate the recognition of just how much work has gone into the processing of this imagery, and our focus on lesser-studied glaciers. We were particularly pleased to read that Reviewer 2 considered our work to be 'an important scientific contribution' and that our 'results deserve to be published', while Reviewer 1 considers that (with some changes) our paper 'can provide an impactful contribution'. We are also appreciative of the thorough reading of our work by both reviewers and yourself, and as a result, they have raised a number of issues and concerns which we now deal with one by one. Each relevant statement from the reviewers is indicated by red text, whereas our responses are in black text.

**Editor**

(E1.1) SCIENTIFIC QUALITY / RIGOR
The methods used to process the 1930s imagery seem appropriate and the length change data appears solid

We are pleased that you approve of our image processing and calculation of length changes. There are two parts to this question that we respond to separately:

although I would like error bars to be included.

We appreciate the need to state uncertainties. We interpret the request for "error bars" to mean the addition of information about uncertainties in the measurements. Raw retreat rates are given for individual glaciers across each time period in Table 2. These retreat rates are visualised graphically in Figures 3, 4 and 5. The distribution of retreat rates for each time period is visualised via box plots in Figure 6. None of these figures are suitable for the addition of graphical error bars so, instead, we now state uncertainties for measurements for each time period in the caption of Table 2. We have also added a description of the source of these uncertainties to the end of Section 2.4. To clarify, the only source of uncertainty in deriving the distances between our glacier front centre positions is the accuracy of the orthomosaics on which the centre point positions are annotated.

Maybe authors could also clarify how a mean/single length change is derived when the shape of the glacier front is complex.

In order to derive a single length change for each glacier, we use a standard single centre-line approach. The centre-line was derived using contemporary velocity data. So, the raw length change data does not involve averaging over any measurements (though note

that in Figures 3-6 and Table 2 we normalise for the duration of each timespan by reporting rates of change in metres per year rather than length change in metres). We have added an explanation of how length changes are derived in Section 2.4.

**(E1.2) SIGNIFICANCE / IMPACT**
Maybe the article would have more impact if a clearer take home message about the drivers of length changes could be proposed. But maybe this is just not possible given the heterogeneity of the observed signal?

As a consequence of the comments of the two reviewers, we have modified and strengthened our conclusions. Reviewer 1 in particular made a number of suggestions to help improve our findings, and having acted on these, we hope this provides a sufficiently clearer take-home-message. To summarise here, we focus on glacier heterogeneity, and stress the importance of shallow ridges as having controls on the rates of retreat of marine-terminating glaciers, and also focus on the substantially lengthened record over which observe changes in retreat rate, of what was previously available. Another key take home is our focus on the urgency of exploring subglacial topography because it is this that we propose has an important control on the variability in retreat rate observed in our study glaciers.

**(E1.3) PRESENTATION QUALITY**
Section 3.5 lies a bit uncomfortably between the results and the discussion sections. Should not authors merge it with the discussion?

This point was raised by Reviewer 1, and our response to their suggestion can be found below as point (R1.16). Here we say that while we appreciate the view that at first sight, this paragraph may be more appropriate in the discussion section, we believe that there is a strong rationale for keeping its current position. This is because this passage is a rigorous summary of how the various datasets have varied over time, and an attempt to identify co-variability or differences. We feel that such summarising is important to help draw out important relationships before launching into the discussions where we try to understand the mechanism and controls on these variables. A slightly longer explanation is provided in our response to Reviewer 1, and we hope that the Editor will approve of our preference.

**Reviewer 1**

*(R1.1) Comment in paragraph 3*
The discussion needs improvement in order to better support the point that "full understanding of the behaviour and response of the Ice Sheet requires us to consider changes that are taking place at the scale of individual glaciers" (abstract). The paper does an excellent job of showing the heterogeneity among glaciers in the data, but the discussion is weak on interpreting this heterogeneity. I understand that, without observations at most of these glaciers (e.g., bathymetry, oceanographic data), it is not possible to draw conclusions about why some glaciers are stable while others retreat. This paper hypothesises that bathymetry and subglacial topography is likely what is stabilising some glaciers, but this has

been shown in previous work (e.g., Catania et al., 2018). The discussion in the paper needs to be strengthened by showing what can be learned from focusing on the lesser-studied glaciers and by extending the record of glacier behavior back in time, beyond what has already been shown in past literature. In other words, the discussion can be improved by focusing on the novel aspects of the archival data. For example, one question that the authors can consider addressing is: if air and sea-surface temperatures exhibit similar anomalies from the long-term baseline in the 1930s and 1940s as they do in the 2000s and 2010s (Figs. 7 and 8), why do marine-terminating and GrIS outlets respond with so much more retreat in the latter period (Fig. 2)? This is a novel question that could be addressed with the archival data of southeast Greenland that could not have been addressed before.

Thank you for this comment and the suggestions. We have acted on all of the suggestions in the subsequent 'editorial comments' which, together, we feel has strengthened our discussions. As mentioned below, we have also toned down some of our interpretation so that it is slightly more speculative, because we lack the data necessary to draw firm conclusions. For example, a lack of subglacial topography and bathymetry data for large regions. Finally, we have tried to express more clearly the importance of our work and the insights it offers. The reviewer is correct that Catania et al. (2018) have already shown the role of bed topography and bathymetry in modulating flow velocities, and we have now made this clear. We would like to highlight that the fact we see such behaviour in our data too is gratifying. We believe, however, that our insights are unique and make an important contribution because firstly we are working in Eastern Greenland (in a previously unstudied area) and we are exploring unstudied and overlooked smaller glaciers. Secondly, we demonstrate the ability of archival imagery to extend the record over which we are able to draw such conclusions - in fact, we triple the duration of such a study, which we believe to be an important step. In answer to the question the reviewer poses, we believe that the increase in retreat in the latter period, despite air and SST temperatures being similar to the earlier period, is a demonstration of just how important bed topography is - i.e. it has the ability to reveal very different retreat behaviour, through modulation of the role of air and sea surface temperatures.

*(R1.2) General comment 1*

One of the conclusions of the study is that it is important to study glacier-to-glacier heterogeneity in order to get a full understanding of the behavior and response of the ice sheet. While I agree with this sentiment, I feel that this paper frames past work incorrectly. For example, lines 478-480 state that "there has been a temptation in the past to differentiate between regions." I disagree that past work that focused on regional aggregations of glacier behavior did so simply because it was tempting to do this. The goal of grouping regionally is to understand whether there are broad patterns to glacier behavior and their response to external forcing. And, while there have been previous studies that have grouped glaciers regionally, there are also studies that have shown significant heterogeneity between neighboring glaciers (e.g., Bjørk et al., 2012; Carr et al., 2017). I have two suggestions: a. Change the wording throughout the manuscript to more accurately state that the goals of the past regional studies was to investigate broad-scale patterns, while this present study focuses on glacier-to-glacier heterogeneity. OR b. Add to the introduction an overview of past studies that have looked at glacier-to-glacier heterogeneity,

explain what they had found, and then go on to describe how this present study builds on that previous work.

Many thanks for this insightful and well-explained concern. The reviewer makes an excellent point, and highlights the fact that it was not clear that we are interested here in glacier-to-glacier heterogeneity. We definitely did not mean to imply that regional aggregation of glaciers was done just because it was tempting to do so (i.e. without any other scientific reason). However, we recognise that this is not clear from our work. As a result, we have modified the manuscript to follow the first of the two alternative suggestions made - i.e. to make it clear that the goals of past regional studies was to investigate broad-scale patterns, while our work focuses on glacier-to-glacier heterogeneity. As a consequence, we make several modifications:

- At the end of paragraph 1 in our 'Study area' section, we state: 'Dividing the ice sheet into discrete regions like this has proved to be a powerful approach for exploring broad-scale patterns.'
- We also modify a key statement in the final paragraph of the conclusions so that it now reads: 'In the past, regional investigations across the Greenland Ice Sheet have been key (e.g. Mouginot et al., 2019, King et al., 2020). This has been important for exploring broad scale regional behaviour and responses. However, our work here, in which we have focused on glacier-to-glacier heterogeneity, shows that within regions there is great complexity, with even adjacent glaciers behaving very differently.'

*(R1.3) General comment 2*
*An estimate of observation errors for SST and SMB should be added to the manuscript. Currently, Figures 7, 8, and 9 show the standard deviation of the data around the mean (gray shading) but this does not take into account errors in each individual measurement. I appreciate that it is difficult to assess the errors on these data but, at the very least, a rough estimate should be incorporated such that the gray shading includes both the scatter in the measurements and the errors on the measurements themselves.*

We appreciate the request and agree that the SMB and SST datasets we used are not described in detail. However, due to the way that these datasets were produced, it's not clear that a meaningful "observation error" can be provided. The two datasets were created by other scholars that we cite (see below). Their lengthy papers give many details about the raw temperature datasets used, sources and magnitudes of error and their mitigation. To summarise those papers: SMB modelling is based on various datasets covering the time period; temperature datasets span from classical, analoge methodological stations to automated to various satellite based measurement techniques. During modelling, those data sets are combined and, where possible, they are cross validated. They can be, and are, often, used for validation of steps of modelling. Due to this, it is not the raw data that is used for final models but data adjusted in the way that it would remove obvious errors at least in overlapping data sets. If the reason for systematic error occurrence is known the whole data set, not only the overlapping part, can be adjusted. This is why it would be hard to estimate individual observation errors. The original data sets might have their errors, for example, derived from device sensitivity, but after cross validation and removing of various systematic errors, this information is no longer valid for a combined data set. Due to all this, various other statistical methods are being used for error analysis like RMSE as it is described in

detail in Box et al. (2009) and Rayner et al. (2003). A good example is Box and Colgan (2013) when, in order to analyse their 1840-2010 SMB results, they used the Monte Carlo method to analyse the marine ice loss parameter used. They found that before 1991 (a point in time when a new mass balance data set was introduced) their uncertainty increases linearly and, after 1991, it is constant. Due to our inability to incorporate observational errors, we do include a level of variation (standard deviation) in our plots in order to incorporate any lag or variability in time (and space). This is a fairly lengthy explanation, and we do not believe that our paper would benefit from having this added. Throughout the relevant parts of our manuscript we extensively refer to the references below, in which details of uncertainties can be found. We hope that the editor and reviewer will deem this acceptable.

- Rayner, N. A., Parker, D. E., Horton, E. B., Folland, C. K., Alexander, L. V., Rowell, D. P., Kent, E. C., and Kaplan, A.: Global analyses of sea surface temperature, sea ice, and night marine air temperature since the late nineteenth century, Journal of Geophysical Research: Atmospheres, 108, https://doi.org/https://doi.org/10.1029/2002JD002670, 2003
- Box, J., Yang, L., Bromwich, D., and Bai, L.-S.: Greenland Ice Sheet Surface Air Temperature Variability: 1840–2007, Journal of Climate J CLIMATE, 22, 4029–4049, https://doi.org/10.1175/2009JCLI2816.1, 2009
- Box, J. E. and Colgan, W.: Greenland Ice Sheet Mass Balance Reconstruction. Part III: Marine Ice Loss and Total Mass Balance (1840–2010), Journal of Climate, 26, 6990 – 7002, https://doi.org/10.1175/JCLI-D-12-00546.1, 2013.
- Wake, L., Huybrechts, P., Box, J., Hanna, E., Janssens, I., and Milne, G.: Surface mass-balance changes of the Greenland ice sheet since 1866, Annals of Glaciology, 50, 178–184, https://doi.org/10.3189/172756409787769636, 2009.
- Hanna, E., Cappelen, J., Fettweis, X., Mernild, S. H., Mote, T. L., Mottram, R., Steffen, K., Ballinger, T. J., and Hall, R. J.: Greenland surface air temperature changes from 1981 to 2019 and implications for ice-sheet melt and mass-balance change, International Journal of Climatology, 41, E1336–E1352, https://doi.org/https://doi.org/10.1002/joc.6771, 2021.

*(R1.4) General comment 3*
*I suggest combining Figures 7, 8, and 9 into one. This would make it easier to compare air temp, SST and SMB.*

This is a good suggestion which we have acted upon. We have also created a mock up of the combined figure as suggested, and show this below. It now makes it much easier to compare how all these variables change, and to spot similarities in behaviour. Note that the x-axes have been modified so that all three plots line up. One consequence of doing this however is that the combined figure caption would be extremely long and would no longer fit on one page along with the new figure. We have therefore reduced the text making up this figure caption to and moved this additional relevant caption-text to the body of the paper, where it is required to fully understand each figure. We hope this is acceptable.

[Figure]

*(R1.5) General comment 4*
Some of the text in the discussion section presents results without any interpretation and the manuscript would be clearer if this text were moved to the results section. I've tried to note this in my editorial comments below but I may have missed some. I suggest the authors go through the manuscript to move any text that does not include interpretation from the discussion section to the results section.

Thank you for raising this point. We have gone through our paper to ensure that we do not do this unnecessarily. There is one exception where we believe there is some merit in retaining some of this description in the discussions. We discuss this in respect of the reviewer's editorial comment relating to lines 430-438 in the original manuscript (below).

*(R1.6) General comment 5*
In general, more explanation is needed throughout the discussion to substantiate some of the interpretation. I've noted this in editorial comments below.

By responding to all of the editorial comments below, we hope that we have now provided sufficient support for our interpretation. We have also tried to tone down some of our thinking, and make it clear that we are being somewhat speculative in our interpretation. We believe it is worth making these speculations though, as they perhaps pave the way for future important work in Greenland.

*(R1.7) General comment 6*
The interpretation of the behavior of Glacier 14 needs to be clarified. First, the text on lines 442-446 states that the bathymetry data in front of Glacier 14 shows that the stable portion of the terminus is located where the bed is deeper. Although the bed may be deeper in front of the terminus, the data doesn't show what the topography is just upstream of the terminus. There could potentially be a steep ridge there and the bathymetry data that's available does show a shallowing towards the terminus. Second, the text on lines 446-449 goes on state that the observed stability is in opposition to the fact that calving rates tend to be higher for termini in deeper water. This is slightly confusing because terminus stability arises from stable calving and melt rates. In other words, the calving rate can be high but as long as it doesn't increase (and melt rates don't increase), the terminus will not retreat (velocity also has to remain steady). So, it isn't the calving rate itself but changes in the calving rate that are important. Finally, this paragraph hypothesizes that the western part of this glacier is "subject to dynamically-driven mass loss" even though the terminus position is stable. This interpretation is not supported and I would like to see more explanation for why the authors think this is the case. The evidence presented is that the western portion has faster ice flow and is grounded in deeper water (although I question this) but I don't see why this leads to the claim that this glacier is dynamically losing mass.

Thank you for these insights. We have now added some text to suggest that there may well be a shallowing in the bathymetry and thus a ridge which pins the westernmost outlet of Glacier 14. We hypothesise that this ridge could be pinning the glacier and thus be the explanation for apparent relative stability here. However, we do still suggest that an alternative explanation is that large amounts of mass loss are still occurring, but these are countered thanks to faster ice flow. We have also toned down our apparent confidence in our interpretation so as to be a little more speculative.

*(R1.8) General comment 7*
Similar to my previous point, the text on line 455 states that Glacier 15 is undergoing "surface thinning" and is "losing the greatest amount of mass" however this is not supported. Please add justification for these claims.

Following on from the changes made and outlined above, we have removed this statement about losing the greatest amount of mass and surface thinning. Instead we comment on the

fact that ice velocities are high and frontal retreat is also high. Following the thinking used in the consideration of Glacier 14, these two things together suggest that this outlet is most likely to be losing most mass.

*(R1.9) General comment 8*
The data availability statement does not comply with the requirements of the journal. The journal data policy states that if the data is not made publicly accessible in a FAIR repository, an explanation needs to be provided for why this is the case (https://www.thecryosphere. net/policies/data_policy.html). My strong recommendation is for the orthophotos from the BAARE and CORONA, as well as the derived terminus positions from all of the imagery (including those from Landsat), be placed in a FAIR repository, with a DOI obtained and referenced in the Data Availability section.

We will update the data availability statement such that it is compliant. For clarity, the source datasets are all available under their own terms. Specifically, the Corona data set has its own DOI: Declassified Satellite Imagery - 1 Digital Object Identifier (DOI) number: /10.5066/F78P5XZM and can be obtained from USGS (United States Geological Survey) via EarthExplorer - some of the images are available for free some are available at the cost of 30 US Dollars. The BAARE expedition dataset has been given to the Scott Polar Institute mostly by descendants of the original members of expeditions or other institutions that held the images and developed/undeveloped glass plates. As such they are under the protection of the Institute and can be used for scientific reasons once an agreement has been reached. To the maximum extent permissible by the terms of our access to the original source datasets, we will make the processed orthophotomaps publicly available in a FAIR repository with DOI. We will also include the data set of shape files of frontal positions between 1985-2019. All data will or is already available at PANGAEA database.

**Editorial Comments**

(R1.10) [line 279] Change "Thss" to "This"

Correction made as suggested.

(R1.11) [lines 308-309] The sentence "Those outlets ... greater variability" repeats what has already been said earlier in the paragraph about marine-terminating glaciers. I suggest removing this sentence.

We do as suggested and remove the sentence. Removing the text that mentions ocean currents, bathymetry etc also makes sense as this is more interpretative and thus not appropriate in the results section.

(R1.12) [lines 309-310] I suggest moving this sentence to the discussion because it does not directly describe results.

We agree (see response to previous comment). We have moved this to the discussions. However, we had some similar text already relating to this statement, but have expanded this slightly. Consequently, at the end of the penultimate paragraph of section 4.1, we now say: 'Such behaviour is well-documented (Moon and Joughin, 2008; Murray et al., 2015), and highlights that the oceans (currents, tides and bathymetry) and SST changes have a vital role in the stability of these ice masses'

(R1.13) [lines 311-312] These sentences should be moved to the paragraph where Figures 2 and 3 are described.

While we understand the rationale for this suggestion, we believe there is a good reason to retain its current position. The sentences prior to this contain considerations of change and variability in our different types of study glacier. However, we also observe that some glaciers show marked stability and we feel that this is an apt way to conclude this section. On this basis, if the reviewer and editor approve, we propose to leave this sentence where it is.

(R1.14) [line 329] I suggest renaming this section to "Surface mass balance"

Correction made as suggested.

(R1.15) [line 365] I suggest changing "controllers" to "external forcings"

Correction made as suggested.

(R1.16) [lines 365-371] This paragraph should be moved to the discussion. It can be combined with the existing text in Section 4.1.

Again, we understand the rationale that this paragraph appears at first sight to be more appropriate in the discussion section, we believe that there is a strong rationale for keeping its current position. This section (3.5) is a rigorous summary of how the various datasets have varied over time, and an attempt to identify co-variability or differences. We feel that such summarising is important to help draw out important relationships before launching into the discussions where we try to understand the mechanism and controls on these variables. Identifying lags (as we do in lines 365-371) is an important part of this, and we feel it would be out of place to have to draw this out within the discussions rather than here in the results (and indeed in a subsection dedicated to summarising all the data).

(R1.17) [lines 380-382] Please add the start and end years that define each of these periods discussed in this sentence

Suggestion made as suggested. In the first paragraph of the discussions, we have modified the text to read: 'Greater rates of retreat take place during the warmer period (approximately

1925-1964), with a more subtle slowing of this response during the cooler periods (approximately 1905-1925 and 1964-1996), and a faster retreat/collapse in the contemporary period (approximately 1996 onwards) (Hanna et al., 2012; Van den Broeke et al., 2016).'

(R1.18) [lines 383-384] Please add some discussion of this differences in air temperature from this study and what was shown by Hanna et al. (2021). Is this just the difference between Greenland-wide air temperature versus the trend in the southeast? Or is there a difference between the air temperature reconstructions for the same region?

There is some complexity here. Firstly, we acknowledge that our statement of increasing temperatures in the contemporary period was perhaps a little simplistic, as there is a slight flattening in recent years, which we now state very briefly in point (iv) of section 3.5. Beyond that, we believe that the interpretation of Hanna et al. (2021) is perhaps more rigorous than our own in the contemporary period. Hanna et al. (2021) state that there are compensating warming and cooling periods that together mean no significant net temperature change (we are not clear from the work of Hanna et al. (2021) whether this lack of change is from 2001 or just from 2012 - this is not clear in their work). We therefore modify our text to reduce the significance of the apparent conflict between Hanna et al. (2021) and us. We now replace:

*Interestingly, our direct observation of increasing air temperatures in the contemporary period somewhat contradicts the recent work of Hanna et al. (2021) who suggest that Greenlandic air temperature trends are generally flat since 2001. Figure 7 suggests a small but important rising temperature trend in this part of Greenland. This is particularly so in the record of minimum air temperature, which may be significant when considering the role of elevated minimum temperatures on the net amount of melt that takes place.*

with:

*Hanna et al. (2021) suggest that Greenlandic air temperature trends are generally flat since 2001. Taking this period in isolation, although there is some clear variability, it is possible that the overall trend is flat or at least subdued. However, taken as a whole, we believe that temperature trends in the contemporary period do show an overall increasing trend. This is particularly so in the record of minimum air temperature, which may be significant when considering the role of elevated minimum temperatures on the net amount of melt that takes place.*

(R1.19) [line 385] Please explain why this increase is seen as important. Was there a threshold crossed? Is it the largest observed increase in XX years? I also suggest mentioning the positive degree days in the contemporary period and discuss their behavior.

Given we have now modified this section, we no longer state that the increase is important and so do not respond further to this point. We do, however, add a small amount of material about the positive degree days. We move the passage

*'In the contemporary period, we also see warmer seas, as well as a larger increase in positive degree days.'*

…so that it is now at the end of that paragraph, and modify it so that it reads:

*'In the contemporary period, we also see warmer seas, as well as a larger increase in positive degree days. There is considerable variability from year to year in the positive degree days during this period, which perhaps reflects the compensating short term warming and cooling events referred to by Hanna et al. (2021)'.*

(R1.20) [lines 391-392] More discussion is needed to explain why the interpretation is that warming waters are responsible for glacier retreat, rather than rising air temperatures.

On reflection, we realise that this statement was somewhat presumptuous. We do not know that warming waters are responsible, but rather we hypothesise that warming waters may well be influential. This is because our data shows that waters have warmed, and so we feel that these must have some role, along the lines suggested by Wood et al. (2021). It is, however, correct, that we do not know this to be the case and so we rephrase the passage. It used to read:

*It is these warming waters that we believe are responsible for the retreat of many of the tidewater glaciers in our sample.*

But we now say:

*We hypothesise that the warming waters that we observe may well play an important role in the mass loss we observe.*

(R1.21) [line 400] Add the word "detail" following "more"

Correction made as suggested.

(R1.22) [lines 402-404] This sentence, which states that SMB is a driver of marine-terminating glacier change in recent years, seems like it contradicts the previous paragraph, which states that ocean temperature has been responsible for the retreat of these glaciers since the 1990s. I suggest combining the previous paragraph with this sentence and clarifying the interpretation of whether one or both drivers are responsible for frontal changes during various periods.

The confusion here arises from the fact that our statement that SMB has become increasingly important relates to local glaciers and land-terminating glaciers, not marine terminating glaciers. With the modification made above in which we now 'hypothesise' that warming waters are important for marine terminating glaciers, we feel that this previous paragraph and the one under discussion are no longer confusing.

(R1.23) [line 405] Remove the hyphen between "local" and "heterogeneity"

Correction made as suggested.

(R1.24) [lines 430-438] This paragraph presents results rather than interpretation. I suggest moving this text from the discussion to the results section.

We agree that some of the text in this passage is quite descriptive, and thus perhaps more akin to a results section. However, we would very much like to keep this text here if the reviewer and editor approve. The reason for this is that throughout the results, we talk about the glaciers and their behaviour more generally, trying to identify shared characteristics. It is only here in the discussions that we introduce the idea of heterogeneity and variability between glaciers. As a result, it seems relevant to briefly describe the nature of the heterogeneity before attempting to explain this. If we were to move this passage to the results, we would need to introduce the idea of glacier-to-glacier heterogeneity before we had uncovered this in our data and interpretation of those data. We feel this would confuse our line of reasoning and make the text more complicated. Hence, we prefer to leave it in its current position but are happy to follow the recommendation if our rationale is not convincing.

(R1.25) [line 430] Please specify: "2-3 times more retreat" ... more than what?

We mean more than the list of glaciers that were discussed in the previous paragraph (i.e. the majority of those shown in Figures 10 and 11). We rephrase the text so that this is now clearer. We now say: 'In addition to these previously discussed outlets in which frontal change is minimal, there are three outlets that in particular show 2-3 times more retreat than these.'

We feel that either is fine, but we have corrected to 'due'.

(R1.27) [lines 439-441] The first sentence needs to be rephrased because I think it incorrectly presents the conclusions of Wood et al. (2021). In fact, the second sentence of this paragraph contradicts the first. I suggest replacing the first two sentences with the following: "Although the behavior of many of Greenland's marine-terminating glaciers has been speed up and mass loss as a consequence of warming ocean waters (Wood et al., 2021), there are glaciers that have exhibited small or no retreat."

We agree and appreciate the suggested alternative. We have thus made this change.

(R1.28) [lines 439-455] This paragraph hypothesizes the response of glaciers in terms of dynamic thinning and links that to glacier ice speed and calving. However, I'm not clearly seeing the link. For example, it is stated that the eastern portion of glacier 15 is sitting on a

shallower bed and has slower ice speed and, therefore, the terminus retreat is due to calving and there is less dynamic thinning. Why would calving driven retreat and slower ice speed necessarily imply that there is relatively less dynamic thinning? This link needs to be explained in greater detail.

In response to earlier comments, this section has now changed and we no longer make reference to dynamic thinning. Our explanations and hypothesising here are now less forthright and more speculative.

(R1.29) [line 443] I believe that "Glacier 15" should be replaced with "Glacier 14" here

Thankyou for this comment, this has been changed.

**Reviewer 2**

(R2.1) It is unclear to me why first a orthomosaic is created after which it is georeferenced to a DEM. This seems like an inaccurate approach. It would result in a more accurate result if GCPs were introduced earlier in the flow, during SfM-processing. Perhaps I am misunderstanding your workflow… Under all circumstances, then I would like to see a work-flow diagram, to be sure that I have understood to process chain correctly.

Thank you for this comment, we agree that the description of the processing pipeline was insufficiently detailed in the original version of the manuscript. We will improve the explanation in the revised version to incorporate the additional details below and add a work-flow diagram (to supplementary material if space in the main paper does not permit).

In fact, GCPs *were* used during the structure-from-motion processing. In detail, our pipeline was as follows. We produced an initial 3D model using Agisoft Metashape without GCPs in order to check if there was enough overlap between images and to analyse the images in terms of their quality. We also produced masks where they were needed: we removed parts that might have changed between images like icebergs on water, photographic plate boundaries, sky and mountains in the distant background. In the case of the 60s images we masked parts of the image that we were not planning to use, due to the size of these images, this procedure significantly reduced the processing time.

When usable images were chosen and masked we placed GCPs on the images with corresponding 3D locations on our reference DEM (ArcticDEM), then we allowed the SfM algorithm to additionally find tie points between the images. At this point we analysed the GCPs. If their reprojection errors were significant we double checked if their placement over the series of images was correct. If so, and the error was still large, we removed them from processing (assuming the error resulted from errors in ArcticDEM described for example in Meddens et al. 2018). After that check we processed tie points again and produced orthomosaics from the 3D models. So, to clarify the specific question: orthomosaics are georeferenced as part of the SfM processing pipeline, not as an independent step afterwards.

 Since you are not producing a DEM, then why are you not using an image source as master for the GCPs? It seems like an inaccurate approach for rectification of an image.

Again, the improved explanation we propose above will help clarify this misunderstanding. Some of our images (specifically from the 1930s BAARE dataset) are highly oblique. An orthomosaic cannot be produced from these images without going via a 3D model to compensate for the significant occlusions and perspective changes to produce a top-down orthomosaic. Hence, we require 3D GCPs for structure-from-motion and cannot rely on calibrated/georeferenced satellite images. In addition, due to the highly variable appearance over the large time spans we work with, image-based features were rarely useful for matching GCPs. Using a reference DEM and 3D GCPs allowed us to use topographic features for GCPs which were more easily identified and matched in our images and reference DEM. Finally, for consistency we wanted to have one unified ground truth model across all datasets and so using ArcticDEM (and overlapping sets of GCPs) for all datasets was the best choice.

(R2.3) Why are you producing a 1985 ortho with GCPs from ArcticDEM, when an ortho already exists with GCPs from in-situ measured points? You are also referenceing the correct paper, Korsgaard et al. 2016, from which the ortho and DEM was published.

Thank you for this comment, which is certainly a valid question. As mentioned above, we wanted to use the same processing pipeline and reference model for all the data sets. Thus instead of using a ready product we processed the images on our own. Also, we wanted to have more freedom in producing the mosaic with textures and pixel size that we could compare to other data and our own processing pipeline allowed for that.

(R2.4) You mention 58 images used in the text but only 30 in the table. You also mention GCPs from SDFE associated with the images - are these the ones you have used?

There were 58 images originally obtained but due to various reasons described above and also lack of coverage (images of sea or 'white-on-white' Greenland ice cap images) we had to limit the number of actually used images.

We do not mention anywhere in the text anything about GCP from SDFE and we did not use either the existing GPS (GNSS) points nor Doppler points as GCP. We used only GCP from ArcticDEM and, as often as possible, we used the same points between all models. The 1930s images covered a relatively small area since the oblique images were taken from low height and the photo cover is not continuous over the shore. Taking all this into account the double coverage of those GPS and Doopler points would have been extremely poor thus we decided to work with a more flexible data set (ArcticDEM) that allowed us to more freely choose GCP positions.

(R2.5) There is no information provided on how you reach the 2D and 3D errors in table1. From the way I understand your processing, I don't see how you can have a 3D error, when

you state that: "For geolocalisation of the orthomosaic….. the ArcticDEM model was used. If you have a georeferenced 3D product (DEM) than it would be very nice to see it included in the manuscript.

We used Agisoft Metashape for the creation of orthomosaics. We use 3D GCPs and thus Metashape produces a 3D error for them. We agree that, in the case of orthomosaics, this 3D error does not carry significant information and will be removed from the next version of the manuscript. The DEM model is being produced in the further steps of this project however due to the complexity of the datasets we are still improving its production. Also we feel that with the amount of information that we already have from the orthomosaic and additional temperature data, showing preliminary results related to a DEM would overcrowd the manuscript and take the attention away from interesting results of already fully processed orthomosaic data.

(R2.6) I would like some more information on the SMB model and specifically the area of the model the results that you are showing here represents. Since it is shown as a point/line graph does it represent the combined glacier area studied or a point in the region? Would be interesting to see the SMB plotted on a map.

The SMB results are taken from the *SE region*; Greenland wide SMB model performed and presented in Box (2013), Box et al. (2013) and Box and Colgan (2013). Since they are the results of a modeling procedure they cannot be considered strictly as observations. The *SE region* is defined by portioning the ice sheets' drainage basins (we have attached a screenshot of the basins we believe are used for this region - highlighted in yellow). According to the mentioned papers, the cross validation with GEUS/ Denmark Meteo institute of 'SE Greenland' meteorological records proves the models of this region to be highly accurate. We do agree that a spatial SMB plot provides an extremely suggestive visual representation of this dataset. However, those plots have already been done in Box (2013), thus this would not add any extra information for our study. We will make explicit reference to these plots in the revised version. The value provided in the graph is an average for this region. We choose that since it allowed for better time step visual correlation of used temperature and mass balance and also better accompanied box plots showing magnitude of change over the different time-steps (fig. 6).

- Box, J. E. and Colgan, W.: Greenland Ice Sheet Mass Balance Reconstruction. Part III: Marine Ice Loss and Total Mass Balance (1840–2010), Journal of Climate, 26, 6990 – 7002, https://doi.org/10.1175/JCLI-D-12-00546.1, 2013.
- Box, J. E.. Greenland Ice Sheet Mass Balance Reconstruction. Part II: Surface Mass Balance (1840–2010), *Journal of Climate*, *26*(18), 6974-6989. Retrieved Jan 4, 2022, https://journals.ametsoc.org/view/journals/clim/26/18/jcli-d-12-00518.1.xml, 2013
- Box, J. E., Cressie, N., Bromwich, D. H., Jung, J., van den Broeke, M., van Angelen, J. H., Forster, R. R., Miège, C., Mosley-Thompson, E., Vinther, B., & McConnell, J. R.. Greenland Ice Sheet Mass Balance Reconstruction. Part I: Net Snow Accumulation (1600–2009), Journal of Climate, 26(11), 3919-3934.https://journals.ametsoc.org/view/journals/clim/26/11/jcli-d-12-00373.1.xml, 2013

(R2.7) It appears that very few of the glaciers studied have data from the 1930s. Table 2 shows only 7, while fig 3 shows 18. How come have you chosen not to focus only on the glaciers that have the long record. I agree that adding more glaciers gives the dataset more value, but I am missing a justification and most importantly some criteria for your selection of additional glaciers.

Thank you for this comment. We agree that the criteria used there are a bit vague however after much deliberation we decided to include more glaciers than only the 30s data set glaciers. This was dictated by two factors: first was the use of the 60s data set from the

CORONA mision. Due to its complexity this dataset is rarely used and has never before been used to produce data for this part of Greenland, thus it is an interesting result on its own. Second, while mostly focusing on the areas covered by 30s data we produced orthomosaics of the surrounding glaciers and then observed that, even with fewer time steps, these interactions are interesting. We observed that neighbouring glaciers react very differently in similar circumstances and thus decided that this is worth exploring. Also in some cases we were missing some time steps - for example we had 30s and 80s data but no 60s data (due to snow/cloud coverage) thus the lack of one time set is not limited to only 30s data. To summarise, taking into account an enormous time sweep that we are covering and the technical capabilities of each era of photography it is understandable that some data will be missing and we did not want to limit our data set even further just in order to unify the time steps.

(R2.8) Fig 6 is great – would it be possible to combine it with fig 7, 8, and 9, for a better overview?

Thank you for this suggestion. As described in our response to reviewer 1, we agree with the suggestion to combine figures 7, 8 and 9 into one figure (and we show a preliminary version of how this combined figure will look). However, it is not possible to also include figure 6 and still fit all of the content onto one page. So we propose to keep this figure separate.

(R2.9) It would be nice to see on a map from where the temperature is coming – both air and SST. There is no information provided from which grid cell you have extracted the SST. I am not sure what is meant by mean annual maximum and minimum temperatures – can you please explain?

Thank you for this comment. Air temperature used in this paper is coming from Tasilaaq weather station (65.60°N, -37.63°E). This is the longest air temperature record in the region thus is the most representative data source. SST temperature is a combination (average) of values taken from the 1 degree grid cells (67°N and - 31°E -32°E) closest to our study region - this information will be added to the text. The mean annual maximum and minimum temperatures (wrt air temperature) are the average (mean) of the maximum and minimum temperatures recorded for that year (annual).

(R2.10) Several places in the text is mentioned mass loss, you are not providing data to support these statements, and can with what presented only describe retreat.

We have removed mention of mass loss where this was based on a reading of retreat rates. However, we retain mentions of mass loss when we are discussing surface mass balance changes.

(R2.11) In your conclusion you write that there is been a temptation to differentiate between region. I suggest you reword this. Subdivision into regions makes perfectly sense, as climate, ocean currents, landscape and geology varies on a regional scale. While there may be variations within the regions, there are plentiful patterns that warrant these subdivisions.

We agree, and reviewer 1 also raised this point (see response to their comment). We made some modifications so that it is clear that we acknowledge the importance of regional studies. We reinforce this point in several locations (see reviewer 1) and finally state in the final paragraph of the conclusions: 'In the past, regional investigations across the Greenland Ice Sheet have been key (e.g. Mouginot et al., 2019, King et al., 2020). This has been important for exploring broad scale regional behaviour and responses. However, our work here, in which we have focused on glacier-to-glacier heterogeneity, shows that within regions there is great complexity, with even adjacent glaciers behaving very differently.'

**Additional changes**
A1) Towards the end of the first paragraph of section 4.2, we list a series of variables that control glacier behaviour. These are listed and labelled a, b, c etc., but previously we had multiple variables listed as 'c'. This has now been corrected so the list reads a, b, c, d, and e.
A2) We have added a sentence to our Acknowledgements section in order to thank the editor and reviewers for their work.

---

## Referee Report (RR1)

**Review of tc-2021-256: Unravelling the long-term, locally-heterogeneous response of Greenland glaciers observed in archival photography**

In this revised manuscript, the authors develop a processing pipeline for archival aerial and satellite images and use these images to create a record of nearly a century of glacier terminus variations in central eastern Greenland. They compare the record of glacier terminus change to climatic datasets (air and sea-surface temperatures and surface mass balance) and some bathymetric data. The results demonstrate substantial heterogeneity in glacier terminus behavior and response to climatic forcings, and the authors suggest that bathymetry may play an important role in pacing glacier retreat.

Processing the archival aerial and satellite imagery is a huge effort and I commend the authors for doing that work to substantially extend the record of glacier change in this region back in time. In addition to that work, I think the main strength of this paper is the exploration of heterogeneous glacier behavior between different glacier types, between neighboring (and climatically similar) glaciers, and through time. As the manuscript has already been through a productive round of review, I have only minor further suggestions for changes before publication.

**Comments:**

- L61: The "data sparseness" mentioned here is nicely illustrated by Goliber et al. preprint in TC (https://tc.copernicus.org/preprints/tc-2021-311/).
- L260: Clarify that these are mean annual max/min temperatures. I was unclear on this until I saw Figure 7. Also, how are the positive degree days calculated?
- 2.4: First there is a description of manually delineated margins, then a description of centerline positions. Clarify, were both methods used, and what was the purpose of each?
- Figure 5/Table 2: In Table 2, southern glaciers 22-24 have advance/retreat listed in 1966-1985, but that is not shown for these glaciers in Figure 5. Also, L301-302 state that the southern glaciers have no data from 1930-1985, which is contradicted by Table 2.
- Figure 6: It took me several minutes to figure out that the plots in the center were showing the same data as those at the left, but with an extended y-axis to show the full error bars. Please clarify this in the figure caption.
- L437: Specify that warming waters could contribute to the mass loss of *marine-terminating* glaciers (not all glaciers in your study).
- L446/generally: Many studies that have looked at the effect of warming ocean temperatures on glacier behavior have assessed temperatures at depth, not just SST. While your data focus on SST, it would be good to mention this as well.
- L458-460: the items listed here would benefit from supporting citations as there has been much work on all of these topics as pertains to glacier response.
- Figures 8, 9: Use a perceptually uniform colormap for velocities (see https://matplotlib.org/stable/tutorials/colors/colormaps.html for Matplotlib examples; see

doi.org/10.1038/s41467-020-19160-7 for reasoning). Also it appears that the colormap is not discretized although the colorbar is, which makes interpretation more difficult.

- L560-567: I think that you state the importance of bed topography too strongly in the conclusions given the highly speculative nature of your discussion. The reference to Catania et al. (2018) is important, but unlike here, that paper specifically focused on the influence of bed topography on glacier behavior, as have others (e.g. Carr et al., 2015; Felikson et al., 2021).
- Generally, this manuscript would also benefit from introduction/discussion of past studies that have also highlighted heterogeneous glacier behavior.
- There are a number of typographical errors throughout the manuscript. I've listed those that I identified below; please review the manuscript for any further changes.
  - L57: ETRS1 → ERTS1 (may also mention that this mission is now known as Landsat-1 to improve name recognition)
  - L70: utilse → utilize
  - L79: "poorly understudied" is redundant; use "poorly studied" or "understudied"
  - L128 and elsewhere: use CORONA consistently
  - L125: lead → led
  - L189 and L214: ArcticDEM
  - L282 and elsewhere: I think, Bjork → Bjørk (per other references in the manuscript)
  - L297 and L303: terminii → termini
  - L334: temperature remain → temperatures remain
  - L529: it's → its; to just to → just to
  - L568: reagion → region

---

## Author Response (AR2)

Dear Editor

Many thanks for overseeing the latest round of reviews of our paper: '*Unravelling the long-term, locally-heterogenous response of Greenland glaciers observed in archival photography*'. We are very pleased to see that Referee #2 (Anders Bjork) is happy with our latest version, but sorry to see that Referee #1 was not able to provide a 2nd review. We are nevertheless grateful to the anonymous third referee.

Referee #3 is generally very positive and we are grateful for their acknowledgement of the 'huge effort' that we have put in to 'extend the record of glacier change in this region back in time'. We are pleased that they also recognise one of the focuses of our paper which is to explore heterogeneous glacier behaviour. As before, we deal with each of the comments made by referee #3 in turn, using red text to identify relevant statements from the reviewer and black text to indicate our response.

**Referee #3**

L61: The "data sparseness" mentioned here is nicely illustrated by Goliber et al. preprint in TC (https://tc.copernicus.org/preprints/tc-2021-311/).

Thanks for directing us towards this work. We have now added reference to this paper.

L260: Clarify that these are mean annual max/min temperatures. I was unclear on this until I saw Figure 7. Also, how are the positive degree days calculated?

We now replace: 'Annual air temperatures (minimum and maximum) as well as positive degree days (calculated from these data)' with: 'Mean annual minimum and maximum air temperatures, as well as positive degree days (calculated from these data)'. Positive degree days were calculated by counting the number of days in a year when the maximum temperature record exceeded zero degrees.

2.4: First there is a description of manually delineated margins, then a description of centerline positions. Clarify, were both methods used, and what was the purpose of each?

We apologise for the lack of clarity here. We manually digitised the margins of each glacier at each time-step and then constructed centrelines using the method outlined in our manuscript. We then looked at where the centreline intersected the frontal margin at each timestep to investigate change. In the text, we replace: 'Specifically, we use contemporary velocity data to define a centre-line and measure a single length change from these centre-line positions.' with 'Specifically, we use contemporary velocity data to define a centre-line and measure a single length change at locations where the centre-line intersects delineated frontal margins.'

Figure 5/Table 2: In Table 2, southern glaciers 22-24 have advance/retreat listed in 1966-1985, but that is not shown for these glaciers in Figure 5. Also, L301-302 state that the southern glaciers have no data from 1930-1985, which is contradicted by Table 2.

Thankyou for this comment, figures 5 and 4 are correct however the table and the text was misleading. For glaciers 22-24 and glaciers 15-16 we do have orthomosaics created based on data from BAARE expedition (1930s) and from 1985 mission, however we do not have data from 1960s Corona mission. The data in table 2 refers to change in the glacier frontal position between 1930-1985. Since we can see how this is confusing in the current version of the manuscript we moved all five glaciers to the bottom of table 2 and added a separate heading for this part of the table. The sentence in L301-302 has been changed to "Indeed we do not have any data for the period of 1930-1985 for glaciers 20 and 21. We are also missing the 1966 time step for glaciers 22-23 where we do have data from 1930s and 1985 for those glaciers, and so parts A and B of Figure 5 are blank."

Figure 6: It took me several minutes to figure out that the plots in the center were showing the same data as those at the left, but with an extended y-axis to show the full error bars. Please clarify this in the figure caption.

You are correct and we apologise for omitting this from our figure caption. This has now been amended.

L437: Specify that warming waters could contribute to the mass loss of marine-terminating glaciers (not all glaciers in your study).

We agree that when we refer to SST, bathymetry and the role of sea ice, we only mean these are relevant for marine terminating glaciers. As a result, we replace the following sentence:
'Such observations suggest that glacier response is defined not only by climatic variables (e.g. air temperature, SST) but also by a) ice velocity (and changes in this over time), b) ocean circulation at a calving front, c) underlying topography (i.e. bed elevation beneath an ice mass) and bathymetry, d) the presence, concentration and role of sea ice, and e) ice thickness.'
With:
'Such observations suggest that glacier response is defined not only by climatic variables (e.g. air temperature, SST) but also by a) ice velocity (and changes in this over time), b) ocean circulation at a calving front, c) underlying topography (i.e. bed elevation beneath an ice mass) and bathymetry, d) the presence, concentration and role of sea ice, and e) ice thickness. Of course, the role of SST, ocean circulation, bathymetry and sea ice are only relevant controls with respect to marine terminating glaciers.'

L446/generally: Many studies that have looked at the effect of warming ocean temperatures on glacier behavior have assessed temperatures at depth, not just SST. While your data focus on SST, it would be good to mention this as well.

Thank you for this suggestion. We now draw attention in our discussions to this fact. We add the following sentence to the end of paragraph 2 of section 4.1:

'It is, however, important to note that the focus of Wood et al. (2021) is on subsurface water temperatures that occurred as a result of the spreading of ocean heat caused by changes in the North Atlantic Oscillation (NAO). We do not have data that enables us to explore subsurface temperatures in this way'.

In the following paragraph, when summarising the controls on glacier frontal position, we now also refer to the role of subsurface temperature changes as well as SST changes.

L458-460: the items listed here would benefit from supporting citations as there has been much work on all of these topics as pertains to glacier response.

We are sorry, but we're not clear what it is that the reviewer wishes us to add references to. This passage (lines 458-460) reads:

'*...show 2-3 times more retreat than these. Two of these are part of Glacier 8 (see Figure 8) which has several outlets. The two southernmost ones showed rapid and large-scale retreat between 1930 and 1966, but then displayed very little change over the years since then. In contrast, Glacier 10 showed relatively modest retreat from 1930 to 2015, but then large-scale and rapid…*'

This passage is a description of the behaviour of our glacier. However, we wonder if the referee means the material on lines 441-445, which currently reads:

'*Such observations suggest that glacier response is defined not only by climatic variables (e.g. air temperature, SST) but also by a) ice velocity (and changes in this over time), b) ocean circulation at a calving front, c) underlying topography (i.e. bed elevation beneath an ice mass) and bathymetry , d) the presence, concentration and role of sea ice, and e) ice thickness. Of course, the role of SST, ocean circulation, bathymetry and sea ice are only relevant controls with respect to marine terminating glaciers*'.

We believe the referee is requesting citations related to each of the points we make here, which we now add. As a result, the passage reads:

'*Such observations suggest that glacier response is defined not only by climatic variables (e.g. air temperature, SST) but also by a) ice velocity (and changes in this over time; King et al., 2020), b) ocean circulation at a calving front (Wood et al., 2018), c) underlying topography (i.e. bed elevation beneath an ice mass) and bathymetry (Catania et al., 2018), d) the presence, concentration and role of sea ice (Carr et al., 2013), and e) ice thickness (Barr et al., 1998). Of course, the role of SST, ocean circulation, bathymetry and sea ice are only relevant controls with respect to marine terminating glaciers*'.

Figures 8, 9: Use a perceptually uniform colormap for velocities (see https://matplotlib.org/stable/tutorials/colors/colormaps.html for Matplotlib examples; see doi.org/10.1038/s41467-020-19160-7 for reasoning). Also it appears that the colormap is not discretized although the colorbar is, which makes interpretation more difficult.

Thankyou for this comment, in this version of the manuscript we used colour maps magma and viridis (https://bids.github.io/colormap) for velocities and bathymetry respectively. And the colour bar is no longer discretized.

L560-567: I think that you state the importance of bed topography too strongly in the conclusions given the highly speculative nature of your discussion. The reference to Catania et al. (2018) is important, but unlike here, that paper specifically focused on the influence of bed topography on glacier behavior, as have others (e.g. Carr et al., 2015; Felikson et al., 2021).

Again, there seems to be some confusion over line-numbering. The line numbers refer to the reference list. However, the referee refers to oru mention of Catania et al. (2018) in the conclusions. We refer to this paper on line 523. Shortly before we mention this paper we state:

'*...we propose that the great variability in the retreat of marine terminating glaciers (both in terms of the magnitude and timing of retreat) is strongly controlled by the presence or lack of shallow ridges which act to pin glaciers as they retreat. We envisage an undulating submarine/subglacial topography which has meant that some glaciers have showed periods of much greater or lesser retreat, and some are apparently stable in their position. Such a situation lends itself to the possibility of future periods of comparatively rapid retreat of glaciers that appear to be stable, and likewise future stabilisation of other glaciers that may currently (or in the past) have shown more significant retreat*'.

We are *assuming* that it is in this passage that the referee feels we are overstating the importance of bed topography. To address this, we change the wording slightly so that the passage now reads:

'*...we propose that the great variability in the retreat of marine terminating glaciers (both in terms of the magnitude and timing of retreat) may be controlled by the presence or lack of shallow ridges which act to pin glaciers as they retreat. In our interpretation, we envisage an undulating submarine/subglacial topography which has meant that some glaciers have showed periods of much greater or lesser retreat, and some are apparently stable in their position. Such a situation, if accurate, would lend itself to the possibility of future periods of comparatively rapid retreat of glaciers that appear to be stable, and likewise future stabilisation of other glaciers that may currently (or in the past) have shown more significant retreat*'.

In relation to our mention of the Catania paper, we accept that the wording used seems to imply that their and our work is similar and thus the two pieces of work are comparable. In order to try and reduce this interpretation, we replace:

*'Catania et al. (2018) showed similar results for western Greenland which gives us greater confidence in our interpretation here'.*

With

*'Catania et al. (2018) also provided such insights for western Greenland and so we have greater confidence in our interpretation here'.*

Generally, this manuscript would also benefit from introduction/discussion of past studies that have also highlighted heterogeneous glacier behavior.
We agree that considering other work that has focussed on heterogeneous behaviour is important. However, as we arrive at this idea through the focus of our work, we would like to refrain from discussing this in the introduction at all. Instead, we modify our discussion. The final paragraph of the discussions previously read:

[revised manuscript text omitted]

In addition, we also modify the final 3 sentences of our conclusions. Previously it read:

*'In our efforts to better understand the complexity of the response of the Greenland Ice Sheet to a warming climate, we propose that it is increasingly important to consider the variability between outlet glaciers because of the variation in responses that we have identified here. In addition, our work has also highlighted how difficult it is to analyse overall glacier response from investigations of frontal variations alone. An important future direction would be to focus on surface elevation change and also to explore the subglacial topography of these outlets to predict likely future 'jumping' periods of retreat, or indeed stabilisation'.*

…but we have now modified it to read:

*'In our efforts to better understand the complexity of the response of the Greenland Ice Sheet to a warming climate, we propose that it is increasingly important to consider the variability between outlet glaciers because of the variation in responses that we have identified here. We also support, and further stress the need for much improved knowledge of fjord geometry, as initially called for by Porter et al. (2018) because of its probable importance in controlling the heterogeneity in glacier behaviour. In addition, our work has also highlighted how difficult it is to analyse overall glacier response from investigations of frontal variations alone. An important future direction would be to focus on surface elevation change and also to explore the subglacial topography of these outlets to predict likely future 'jumping' periods of retreat, or indeed stabilisation'.*

There are a number of typographical errors throughout the manuscript. I've listed those that I identified below; please review the manuscript for any further changes.

• L57: ETRS1 → ERTS1 (may also mention that this mission is now known as Landsat-1 to improve name recognition)

Sentence now changed to read: '*Since the launch of the first Earth Resources Technology Satellite (ERTS1;now known as Landsat-1) in 1972 (Ives, 2011)...*'

• L70: utilse → utilize

Corrected as suggested. We actually find three such occurrences in our manuscript, and so we modify them all.

• L79: "poorly understudied" is redundant; use "poorly studied" or "understudied"

We modify to 'poorly studied'.

• L128 and elsewhere: use CORONA consistently

All occurrences of Corona changed to CORONA.

• L125: lead → led

Corrected.

• L189 and L214: ArcticDEM

Corrected.

• L282 and elsewhere: I think, Bjork → Bjørk (per other references in the manuscript)

Corrected - we identify 4 incorrect spellings.

• L297 and L303: terminii → termini

Corrected.

• L334: temperature remain → temperatures remain

Corrected.

• L529: it's → its; to just to → just to

We cannot locate the occurrence of 'it's', but we do make the second suggested correction.

• L568: reagion → region

Corrected.

---

## Author Response (AR3)

Dear Etienne

Thank you for your latest assessment of our paper: 'Unravelling the long-term, locally-heterogenous response of Greenland glaciers observed in archival photography'. We are very pleased to see that our paper has been accepted, subject to us making some final minor changes. We have now acted upon these, and detail how we have responded to these below. Your comments are indicated in red text, with the line-number on which each comment is found in the previous version of the manuscript. How we have dealt with and responded to each comment then follows in black text.

You also mentioned a new paper that we could perhaps reference in our paper. However, we are a bit reluctant to do so for fear of not doing it justice, and in case it appears as if we have added it in at the last minute. We hope that is acceptable.

We hope that you will find our responses to each point below satisfactory.

Kind regards

David

**Line 16**: To which time period does this value refer to? A single mass balance year? Is not 375 Gt/yr sufficient?

We agree that we do not need to state this sea level rise equivalent, so we have removed this as proposed. This opening passage now reads:

"Two decades ago, the GrIS was considered to exist in a state of quasi-stability with its regional climate, but recent climatic warming trends have resulted in it becoming by far the largest contributor to global sea level rise (Hanna et al., 2012; Van den Broeke et al., 2016). Between 1992 and 2018, (3902 ± 342) Gt of ice was lost from the GrIS (Shepherd and IMBIE Team, 2020), but this has accelerated to an annual loss of 375 Gt $a^{-1}$ of ice (on average) in the last decade (Enderlin et al., 2014; Van den Broeke et al., 2016)."

**Line 19**: We are now in 2022, the sentence does not work.

This is a legacy of when we first started working on this paper. However, we have now modified this sentence so that it now reads:

"In 2021, surface melting across large parts of the southern and coastal regions of the GrIS was observed, with 2021 being the joint 14th highest melt year to date, with volumes substantially greater than the 1981-2021 average (http://nsidc.org/greenland-today/)."

**Line 47**: I think the work by Haeberli refers to glacier in the sense of glaciers and ice caps, disconnected from the ice sheets (and inventoried on the RGI). Here you only deal with outlet glaciers of the ice sheet (am I right?). Make sure you do not create any confusion between these two entities. I feel the whole paragraph below does not excatly distinguish the two. There is a bit of room for clarification.

update : I see this is clarified later in the text. Maybe mention here that your focus is on both outlet glaciers and glaciers disconnected from the ice sheet?

You are correct that we are interested in both outlet glaciers and glaciers that are not connected to the ice sheet (although as stated later on in our text: "*The vast majority of the glaciers in our study are outlets of the GrIS but… a small number are identified as being smaller local glaciers and ice caps (GICs) peripheral to the margins of the main GrIS*". On that basis, we believe it is acceptable to retain the Haeberli (2000) reference. We are, however, a bit reluctant to mention at this point that we are exploring both outlet and disconnected glaciers. In the paragraph here in the introduction, we are simply describing how glaciers and ice masses change in response to climate and the fact that there is often substantial variability between ice masses. We feel that it would be out of place to state at this point what our precise interest is, since we do this in the subsection entitled 'Study area'. We feel this is the correct place to make it clear where our interest lies.

**Line 48**: Maybe the 2019 SROCC IPCC report is the best reference here?

We agree and have changed the reference as suggested.

**Line 49**: To be updated.

According to the IPCC's website, AR6 is still to be cited as 'in press'. See: https://www.ipcc.ch/report/ar6/wg2/about/how-to-cite-this-report/

**Line 95**: It seems a lot to me for an individual glacier. Maybe you can double check? Also if this is the discharge, should not you use a positive sign. Make sure you do not describe mass balance values here for which a negative sign would be relevant.

You are absolutely correct and we can only apologise for this error. We are deriving these data from the 'Dataset_S02' document associated with the cited Mouginot et al. (2019) paper. However, we incorrectly have quoted from the datasheet that details cumulative mass balance data. We have thus corrected this and now quote averaged MB data over the period 2000-2017 for both Kangerlussuaq and Helheim glaciers. The text now reads:

"(e.g. Kangerlussuaq (mean mass balance over the period 2000-2017 of −8.52 Gt a$^{-1}$; Mouginot et al. (2019)) and Helheim (mean mass balance over the period 2000-2017 of −6.4 Gt a$^{-1}$; Mouginot et al. (2019)))".

Line 318: did you clearly define that "local glaciers" are the one from Rastner et al. Would be worth doing so already and clearly in the introduction and then stick to this terminology.

Yes, on lines 99-100 of the previous version of our manuscript, we stated:

"The vast majority of the glaciers in our study are outlets of the GrIS but as shown in Figure 1, a small number are identified as being smaller local glaciers and ice caps (GICs) peripheral to the margins of the main GrIS (Rastner et al., 2012)".